# One-step formation of polymorphous sperm-like microswimmers by vortex turbulence-assisted microfluidics

Rong Tan [1,5], Xiong Yang[1,5], Haojian Lu [2,3] & Yajing Shen [1,4] ✉

Microswimmers are considered promising candidates for active cargo delivery to benefit a wide spectrum of biomedical applications. Yet, big challenges still remain in designing the microswimmers with effective propelling, desirable loading and adaptive releasing abilities all in one. Inspired by the morphology and biofunction of spermatozoa, we report a one-step formation strategy of polymorphous sperm-like magnetic microswimmers (PSMs) by developing a vortex turbulence-assisted microfluidics (VTAM) platform. The fabricated PSM is biodegradable with a core-shell head and flexible tail, and their morphology can be adjusted by vortex flow rotation speed and calcium chloride solution concentration. Benefiting from the sperm-like design, our PSM exhibits both effective motion ability under remote mag/netic actuation and protective encapsulation ability for material loading. Further, it can also realize the stable sustain release after alginate-chitosan-alginate (ACA) layer coating modification. This research proposes and verifies a new strategy for the sperm-like microswimmer construction, offering an alternative solution for the target delivery of diverse drugs and biologics for future biomedical treatment. Moreover, the proposed VTAM could also be a general method for other sophisticated polymorphous structures fabrication that isn't achievable by conventional laminar flow.

Spermatozoa, as the active carrier for genetic material from the vagina to the fallopian tube, owes both effective motion and cargo loading ability enabled by its unique structure, i.e., a flexible tail for propulsion and a core-shell head for loading[1]. Inspired by the motion mechanism of sperm (as well as bacterial), a wide variety of micro swimming robots with slender flagellum, flexible tail, or helical microstructures have been proposed relying on diverse actuation strategies, such as chemical, magnetic, acoustic or biological methods, etc[2–4]. Particularly, considering the superiorities of magnetic field in intense penetration, biocompatibility, and remote controllability, magnetic driven based microstructures have been widely adopted as robots for the realization of effective swimming at the small scale[5–7]. Further, to tackle the challenge of loading, scientists proposed some approaches to further functionalize the micro swimming robots, mainly including surface coating[8,9] and internal doping[2,10]. Despite these methods allowing the robot to blend some materials, they can neither ensure the drug actives unpolluted by harsh ambient environments nor eliminate the possibility of the reaction between medicine and robot's materials[11]. Moreover, to avoid the incompliance of patient from abrupt release, the controllable sustained release of drug should also be taken into account during the micro swimming robot design.

[1]Department of Electronic and Computer Engineering, The Hong Kong University of Science and Technology, Clear Water Bay, Hong Kong, China. [2]State Key Laboratory of Industrial Control and Technology, Zhejiang University, Hangzhou 310027, China. [3]Institute of Cyber-Systems and Control, the Department of Control Science and Engineering, Zhejiang University, Hangzhou 310027, China. [4]Center for Smart Manufacturing, The Hong Kong University of Science and Technology, Clear Water Bay, Hong Kong, China. [5]These authors contributed equally: Rong Tan, Xiong Yang. ✉e-mail: eeyajing@ust.hk

Nowadays, it has been revealed that the genetic materials are encapsulated in sperm's core-shell head and protected by the biomembrane, i.e., a thin sperm plasma membrane. Such semipermeable membrane not only protects vulnerable from free radical attack and the generation of oxidative DNA damage, but also maintains the inside-outside membrane ionic interaction[12,13]. With the rapid development of micro/nano technology, scientists have proposed many ways to develop artificial biocompatible membranes, such as electrochemical deposition[14,15], stereolithography printing[16], microfluidic emulsification[17,18], etc. Among all, microfluidic emulsification is regarded as a promising approach with the advances of rapid and reliable monodispersed, steadily uniform, and high throughput production of carriers with membrane. To date, variety of spherical structures with artificial biocompatible membranes (e.g., liposome, polyelectrolyte microcapsule, etc.) have been successfully achieved through utilizing varied microfluidics techniques like co-axial microfluidic[19,20], ferrofluid droplet-templated microfluidic[21], flow lithography microfluidic[22], etc, and verified their capabilities as cell carrier, drug reactor and cargo container. However, limited to surface tension, conventional droplet-templated microfluidics are mostly used in the manufacture of symmetrical and regular microspheres. Despite their excellent loading and protecting ability, the swimming efficiency and controllability of a completely symmetrical structure similar to a sphere in a liquid environment is much lower than that of a flagellum-like swing drive.

In this work, we propose a one-step formation strategy to construct the polymorphous sperm-like magnetic microswimmers (PSMs) by developing a vortex turbulence-assisted microfluidics (VTAM). The developed PSM consists of a flexible tail and core-shell body structure, enabling both controllable propelling and effective loading simultaneously. Moreover, by coating alginate-chitosan-alginate (ACA) layer on the PSMs, stable sustain release can also be achieved, which may find wide application in targeted drug delivery in the future. This research also provides a convenient and quick strategy for fabricating the sophisticated 3D polymorphous structure, which cannot be achieved by the conventional laminar flow device.

## Results

### Design and fabrication of PSMs

To endow microswimmer both loading and propulsion abilities at a low Reynold environment, we propose an asymmetrical design with polymorphous head and flexible tail inspired by spermatozoa's morphology and function. Such sperm-liked microswimmer is fabricated by a VTAM platform, which consists of a cross-shaped microfluidic chip and a vortex container connected by a round glass capillary (Fig. 1 and Supplementary Fig. 1). Firstly, the rudiment is rendered as a monodispersed magnetic alginate/oil droplet containing the to-be-delivered materials. Then, a slender tail is drawn out from the sphere droplet to form the sperm-liked asymmetrical structure for propulsion. Benefited from the continuous microfluidic fabrication process, the flexible tail and polymorphous head can be constructed as an integrated unit with reliable connection. From the view of the formation mechanism, the monodispersed droplet's revolution can be divided into three stages, i.e., encapsulated droplet production, slender tail extraction, and sperm-liked morphology solidification.

In the first stage, a cross-shaped microfluidic chip is developed to fabricate the monodispersed magnetic alginate/oil droplet with uniform shape and size. As illustrated in Fig. 1a, the disperse phase solution (1 wt% alginate solution mix with $Fe_3O_4$ nanoparticles (20 wt%; average size, 230 nm) containing model drug (e.g., FITC)) is introduced into the microfluidic chip by inlet one. Simultaneously, the continuous phase solution (paraffin oil mix with Span 80 (2–2.5 wt%)) is introduced into the microfluidic chip by inlet two. These two solutions meet at the cross-shaped channel (Fig. 1b), and the disperse phase fluid forms the droplet under the shear and squeeze action of the continuous phase. The formation quality of the droplet is relevant

to the channel's width and the two-phase flow rate ratio. In detail, the uniformity of the droplet cannot be maintained if the channel is too wide or too narrow, because too wide channel will make the droplet oversize, while too narrow channel will cause the cross-shaped channel blockage to destroy the droplet homogeneity (Supplementary Fig. 2). Further, to maintain the consistency of the droplet size, the two-phase flow rate ratio should be set in the range of 2.4–3, otherwise the droplet size might have a large variance due to coalescence occurring when droplets collide downstream of the chip. (Supplementary Fig. 3). With the above considerations, we finally optimize the maximum width channel to 50 μm, the narrow channel at the junction to 25 μm and the flow rate to disperse: continuous = 0.5:1.5 μl min⁻¹. Under this condition, the monodisperse magnetic alginate droplets (outer: oil layer, inner: magnetic alginate solution) are formed with diameter ~35–40 μm, which is a conducive size for both effective locomotion and loading.

In the second stage, the formed droplets are transferred into a vortex container (diameter of 1 cm; height of 2.5 cm) filled with calcium chloride solution (1–2 wt%) through a round glass capillary as shown in Fig. 1c. The capillary is vertically fixed 2.5 mm away from the container wall, and a rotational magnetic stirrer (diameter of 5 mm; height of 8 mm, speed of 600 rpm–1400 rpm) is put in the container to generate the vortex. At the nozzle of the capillary, the droplet's oil layer bursts process under the action of vortex flow impact. Then, the inner components (magnetic alginate solution) of the droplet are exposed to the calcium chloride solution directly, and it is extracted along the flow direction caused by high speed vortex flow of the calcium chloride solution. Obviously, the length of the tail is positively related to the flow speed, and a higher flow speed will extract a longer tail.

The third stage, PSMs solidification, happens in a few milliseconds right after the tail is extracted due to the tight cross-linking reaction between the deformed droplet without oil protection and the calcium ions solution (Fig. 1d, e). In this process, the solidification speed, which determines the final structure of the microswimmer, is relevant to the concentration of calcium ions in the solution. Higher $Ca^{2+}$ concentration leads to quicker solidification, resulting in the microswimmers gels immediately before it is extracted by the vortex flow. In contrast, when it comes to low calcium ions concentration, the microswimmers will appear irregular shapes after deformation and extraction due to slowly gelling.

It's worth noting that the droplet is aways subjected to a vortex flow during the demulsification and solidification process. Therefore, the microswimmer can be formed with abnormal curved structure. Meanwhile, by adjusting the calcium chloride concentration and rotation speed, the distorted magnetic alginate are solidified to form diverse polymorphous shape, classified as $\Pi_1$ (helix), $\Pi_2$ (regular) and $\Pi_3$ (irregular), as illustrated in Fig. 1f.

### The formation of polymorphous tailed-magnetic microswimmers

To illuminate the PSMs formation process and principle, we theoretically analyze the alginate/oil droplet evolution mechanism under the action of VTAM platform, including formation, demulsification and solidification. The formation of the magnetic alginate/oil droplets (stage I) can be depicted by Lattice Boltzmann method[23] in cross-shaped microfluidics. Particularly, the thickness of the alginate-oil interface $\lambda$ can be calculated by the Ginzburg-landau free energy function based thermodynamics[24] as $\lambda = 4b\kappa/3\sigma$, where $b$ is the constant coefficient, $\kappa$ is surface curvature relating to the interfacial tension $\sigma$ (Fig. 2a and Supplementary Note 1). This parameter plays a pivotal role in the demulsification process during fabrication. Our analysis reveals that $\lambda$ should be designed negligible and small enough to ensure the demulsification in the generated vortex.

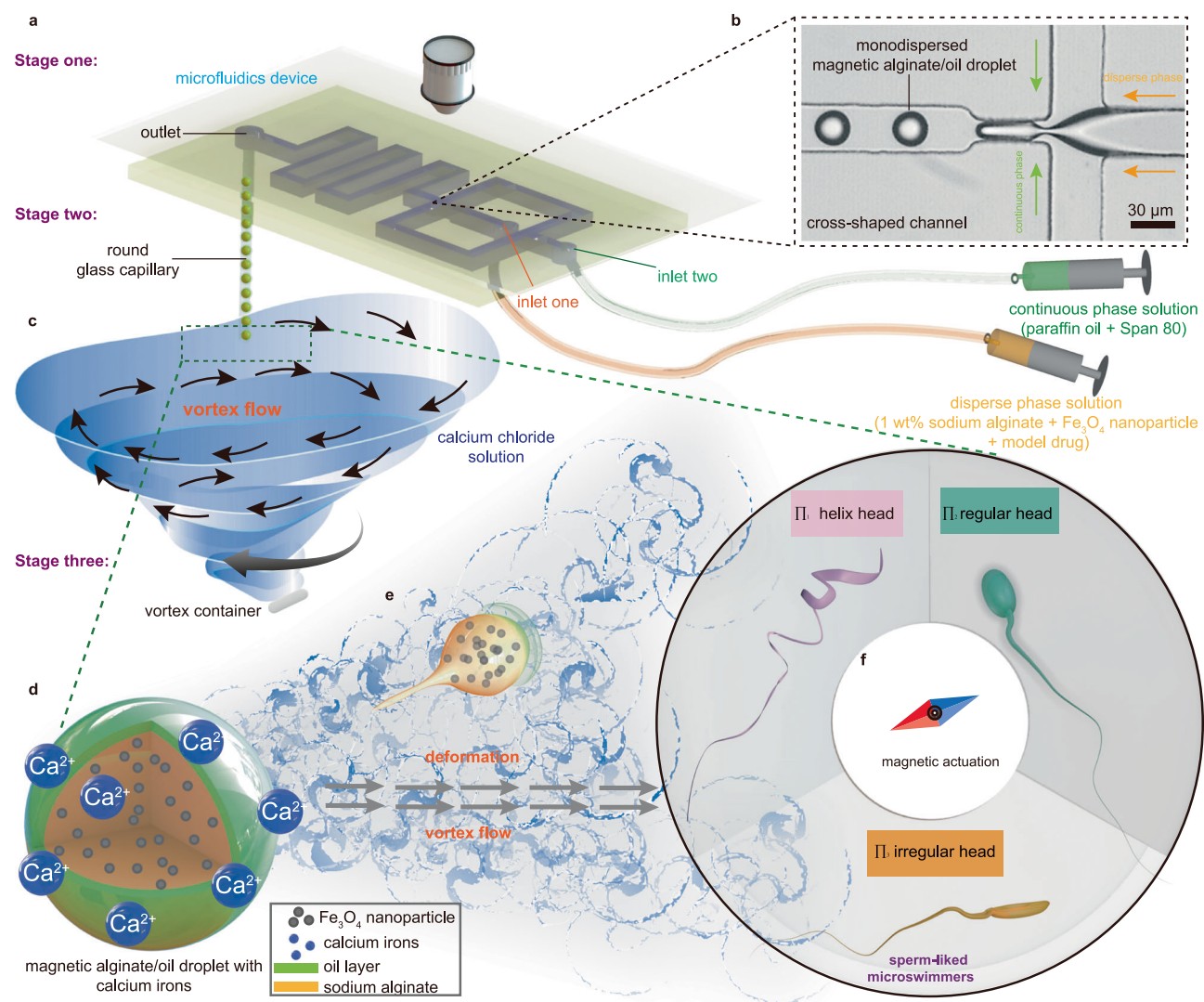

**Fig. 1 | Illustration of polymorphous sperm-liked microswimmers (PSMs) production by the VTAM. a** Chip-based microfluidics consisting of a simple PDMS and a glass slide enables in control alginate/oil droplet, where it is transported into a vortex calcium chloride solution by a glass capillary. **b** The enlarged image of cross-shaped channel for droplet formation. **c** A round glass capillary as a connector, led out from the chip outlet, and placed vertically in a vortex container filled with calcium chloride solution. The droplet flows along the capillary inner and drops into the vortex turbulence environment for suffering fluid impacts from all directions. **d, e** Uncalcified droplets are broken and deformed by vortex flow, and the alginate chains undergo a tight cross-linking reaction with calcium ions. **f** During the process of the gelation, vortex flow assisted the formation of polymorphous sperm-liked microswimmers, including $\Pi_1$: hellix head, $\Pi_2$: regular head, $\Pi_3$: irregular head.

---

To ensure the demulsification process is generated smoothly, the alginate-oil thickness $\lambda$ are adjusted to be negligible small ($\leq 0.01$ droplet radius) due to limitations in achieving higher vortex strengths in experiment. In this case, the droplet deformation gradient can be given as $\boldsymbol{F}_d(\boldsymbol{X}_d, t_d) = \frac{\partial x_d(x_d, t_d)}{\partial \boldsymbol{X}_d}$, where $\boldsymbol{X}_d$ is a material point of the unstressed state, and $\boldsymbol{x}_d$ is the position vector after deformation. The demulsification process comes up when the deformation reaches the critical value (Fig. 2b), which can be determined by two parameters according to viscosity ratio of Boussinesq–Scriven constitutive law, i.e., viscosity ratio of the interior droplet to exterior $\lambda_d$ and capillary number value $Ca = \frac{\mu_d \nu_{\nu e}}{E_0 d_d}$, where $\mu_d$, $\nu_{\nu e}$, $E_0$ and $d_d$ are the dynamic viscosity of oil membrane, the shear velocity, oil membrane surface elastic modulus and the droplet diameter respectively (Supplementary Note 2). As $\lambda_d$ is constant, the capillary number value $Ca$ plays critical role in determining the alginate-oil droplet state, whose critical capillary value is defined as $\Omega_b$. The alginate-oil droplet remains steady

when $Ca < \Omega_b$ and burst when $Ca \geq \Omega_b$. The demulsification process of the hydrogel-oil droplets in low Reynolds coefficient turbulence is also simulated by COMSOL (Supplementary Note 3, see Supplementary Movie 1). Under the flow impact pressure, the demulsification occurs in a transient manner, where the crevasse appearing in an ultra-thin oil membrane is observed at 1 millisecond moment (state II).

After the demulsification, the inner components of the droplet are exposed to the calcium chloride, the solidification process starts (stage III). During this process, the formed microswimmers structure could be diverse, corresponding to different the inner alginate droplets suffer radial velocity ($\nu_{\nu r}$), tangential velocity ($\nu_{\nu t}$), and axial velocity ($\nu_{\nu z}$) and exert deformation in the vortex flow (Fig. 2c and Supplementary Note 4). Because of the vortex flow generated by magnetic stirrer, the dimensionless formed structure $\Pi_{\Omega_s}$ obeys Buckingham's $\Pi$-theorem $\Pi_{\Omega_s} = fct(\Pi_1, \Pi_2, \Pi_3)$, where $\Pi_1 = \sqrt[5]{\mu_h{}^5/\alpha_s Q_s{}^3}$, $\Pi_2 = \sqrt[4]{\mu_h Q_s/\alpha_s d_s{}^4}$, $\Pi_2 = \sigma_s d_s{}^4/\rho_s \mu_h Q_s$ are the dimensionless

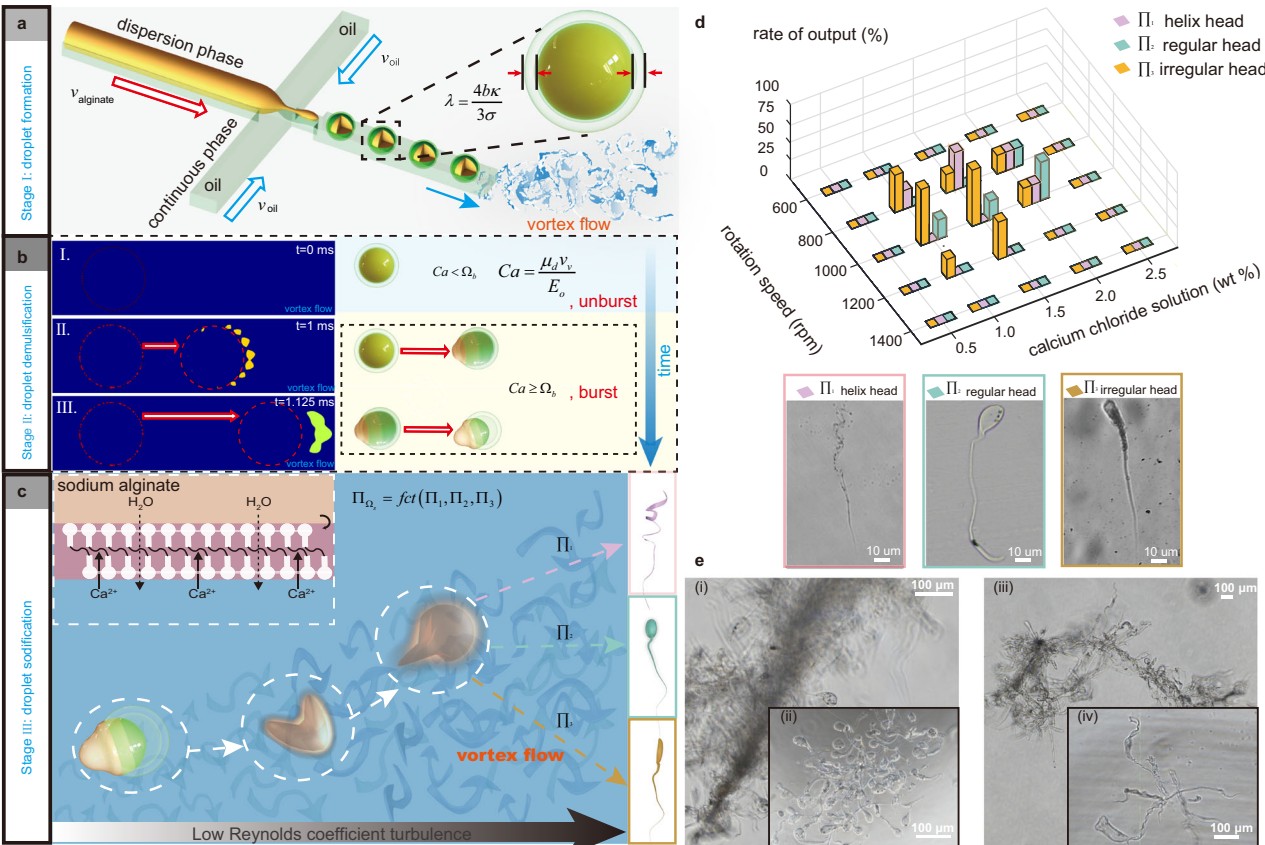

**Fig. 2 | Physical principles of PSMs formation. a** Schematic representation of alginate/oil droplet formation. The different thickness of hydrogel-oil is defined by $\lambda = 4b\kappa/3\sigma$. The alginate/oil droplet are fabricated by cross-shaped channel. **b** Schematic representation of the demulsification stage starts on alginate/oil droplets. A COMSOL simulation of the demulsification process of the hydrogel-oil droplets in low Reynolds coefficient turbulence (Supplementary Movie 1). (I) A critical capillary value $\Omega_b$ is defined to describe the hydrogel-oil droplet state, which is remain steady when $Ca < \Omega_b$ and burst when $Ca \geq \Omega_b$. (II) The gap

appearing in an ultra-thin oil membrane is observed at 1 millisecond moment. Then, with the increase of flow strength, the droplet deformation can be observed in the instantaneous environment. (III) The inside hydrogel phase outflows from fragmentized oil membrane. **c** The hydrogel phase could be solidified to form PSMs by cross-linking reaction. **d** The probability of PSMs productive formation by adjusting vortex flow rotation speed and calcium chloride solution concentration. The morphology of PSMs. **e** The mass production of PSMs.

parameters[25]. Therefore, the structure formation $\Omega_s$ depends on interfacial tension $\sigma_s$, the flow acceleration at the demulsification position $\alpha_s$, extension volumetric rate $Q_s$ with tip diameter $d_s$, fluid density $\rho_s$, and kinematic viscosity $\mu_h$, where the flow acceleration $\alpha_s$ (magnetic stirrer rotation speed) and kinematic viscosity $\mu_h$ (the concentration of calcium chloride) are adopted as regulation parameters.

Through the experimental results manifested in Fig. 2d, two conditions are required for microswimmer productive formation, i.e., the magnetic stirrer's speed should be located in 800–1200 rpm and the concentration of calcium chloride should be 1–2 wt%. Under the condition of 1000 rpm rotation speed and 1.0%/1.5% calcium chloride solution concentration, the production rate of irregular head microswimmers (with head cross-sectional diameter ranging from 5 μm to 8 μm and length ranging from 80 μm to 110 μm) can reach 75% (Fig. 2e, i, iii). While the production rate of regular head microswimmers (with head cross-sectional ranging from 10 μm to 15 μm and length ranging from 90 μm to 120 μm) can reach 50% under the condition of 1000 rpm rotation speed and 2% calcium chloride solution concentration (Fig. 2e, ii). Specifically, the production rate of helix head microswimmer (with head cross-sectional ranging from 2 μm to 4 μm and length ranging from 100 μm to 120 μm) can reach 50% when rotation speed is 800 rpm and calcium chloride solution concentration is 1.5% (Fig. 2e, iv).

## Effect of flow speed/Ca²⁺ concentration on polymorphous tailed-magnetic microswimmers formation

The microswimmers formation process primarily lies in the alginate/oil droplet evolution, encompassing crucial stages like formation, demulsification, and solidification. During the demulsification process, under fluid impact, the droplets detach from the oil film and undergo deformation and elongation to form a tail structure. Simultaneously, a cross-linking reaction takes place with sufficient calcium ions until they are ultimately transformed into microswimmers. According to theoretical modeling, we have obtained the key parameters impacting the fabrication process. These parameters include shear velocity $v_{ve}$, droplet diameter $d_d$, alginate-oil interface $\lambda$, and kinematic viscosity $\mu_h$, with the latter three maintained as constants during fabrication. The principal influencing factor shaping microswimmer structures, as derived from our theoretical model, is the shear velocity $v_{ve}$, which is determined by the vortex strength (Supplementary Note 5). Substituting the known parameters into this simulation elucidates that microswimmers with tails can be formed when the shear velocity falls within the range of 0.6 m s⁻¹ to 1 m s⁻¹ (corresponding to speeds of 800–1000 rpm) (Fig. 3a). In the experiment, when we increase the rotation speed (from 800 to 1000 rpm) we can clearly observe the increase of microswimmers tail, as shown in Fig. 3b. These findings have been verified through subsequent simulations (Supplementary Note 3). The simulations further validate the theoretical insights, providing a comprehensive assessment of the parameter-space

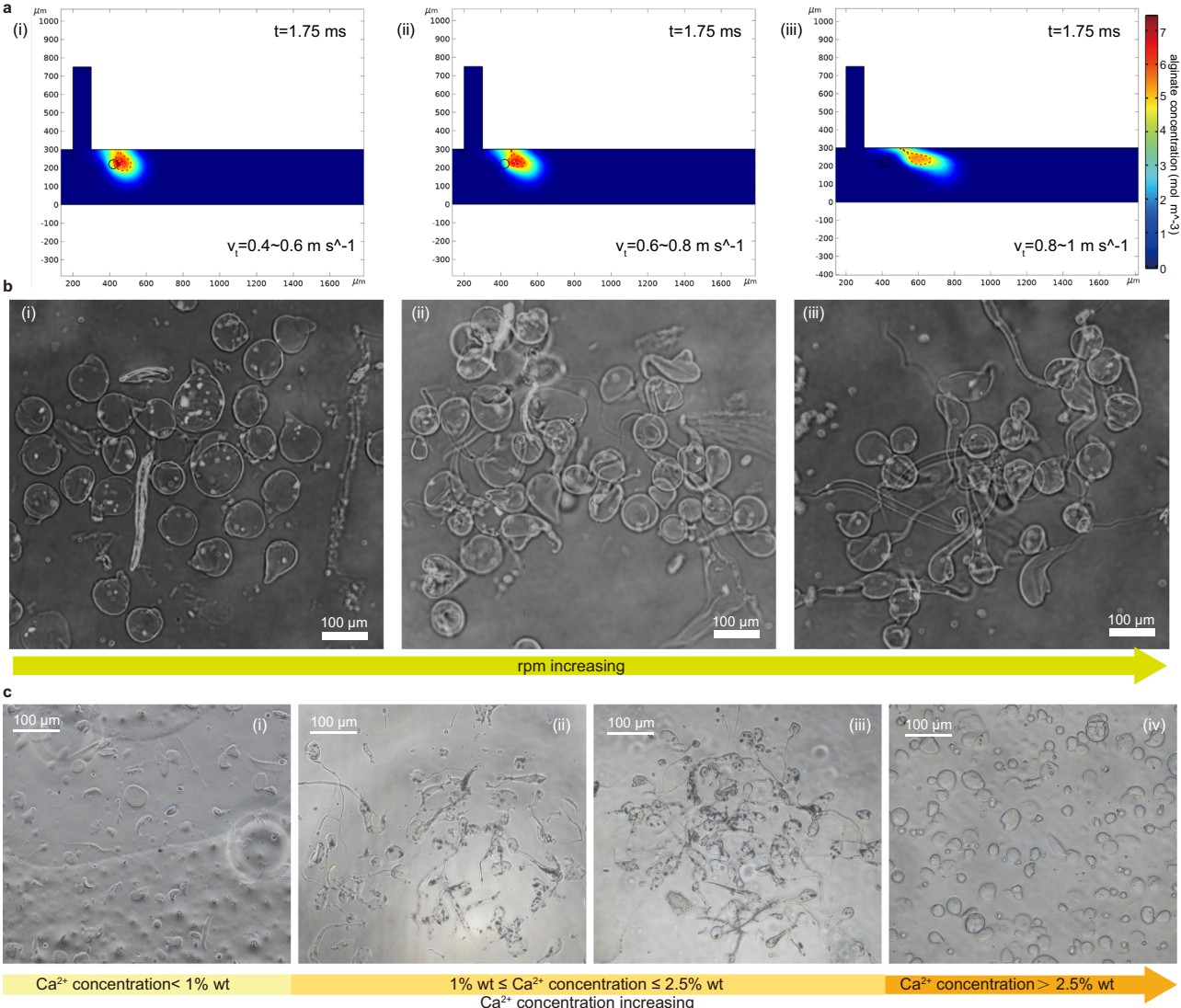

**Fig. 3 | Effect of flow speed/Ca²⁺ concentration on polymorphous tailed-magnetic microswimmers formation.** The comparison of regular head microswimmer's morphology between simulated and experimental conditions at rotational speeds ranging from 800 to 1000 rpm. **a** The simulated results. **b** The experimental results. **c** The observation of microswimmers generated by the VTAM platform under different calcium ion concentrations.

exploration, affirming the influence of shear velocity on the fabrication process of microswimmers.

By observing the experiments, when all conditions remain unchanged except for the concentration of calcium ions (Fig. 3c), we find the concentration of Ca²⁺ affects the solidification process, i.e., when Ca²⁺ concentration below 1 wt% or exceeding 2.5 wt%, the tail-shaped microswimmer cannot be formed. When the concentration is 1 wt% to 2.5 wt%, it doesn't significantly influence the length of the tail (Fig. 3c, ii, iii).

To further validate the effect of Ca²⁺ concentration on gelling speed, we have conducted the comparison experiments as detailed in Methods. As depicted in Supplementary Fig 6, (i) at <1 wt% CaCl₂ concentration, no cap gels formed after 60 min due to insufficient calcium available to bind carboxyl-sites of alginate. The results align with prior studies indicating that lower concentrations of calcium solution hinder gel formation (unable to form a tail-shaped microswimmer, as seen in Fig. 3c, i). (ii) When the Ca²⁺ concentration is high (>2.5 wt%), the microswimmer gels immediately before it is impacted and extracted by the flow. This rapid solidification, induced by high Ca²⁺ concentrations upon contact with sodium alginate droplets, results in round or nearly round microswimmers (as seen in Fig. 3c, iv).

(iii) Within the range of 1–2.5 wt% CaCl₂, there almost no difference in gel morphology. These comparison results indicate that the concentration of calcium ions minimally impacts the formation process. In short, the Ca²⁺ concentrations within the 1–2 wt% range yield a higher number of microswimmers at consistent rotation speeds.

Importantly, our findings underscore the direct influence of flow velocity on tail length, while also acknowledging the role of calcium ion concentration in microswimmer generation. It suggests that, along with the theoretical analysis and simulation results, three types of microswimmer can be manufactured under the appropriate parameters, but to achieve accurate control of the regular swimmer tail's length, the rotation speed can be adjusted.

## Actuation and motion control of microswimmers

Rotation and swing are two general propulsion strategies adopted by microorganisms, because such motions can effectively break the symmetry of the deformation in time to achieve propulsion at low Reynold environment[26]. In our design, we adopt a similar propulsion strategy of sperm in nature, that rotates its body to generate 3D helical wave in its slender tail for propulsion[27]. As illustrated in Fig. 4a, the easy magnetization axis **M** of the asymmetric head of

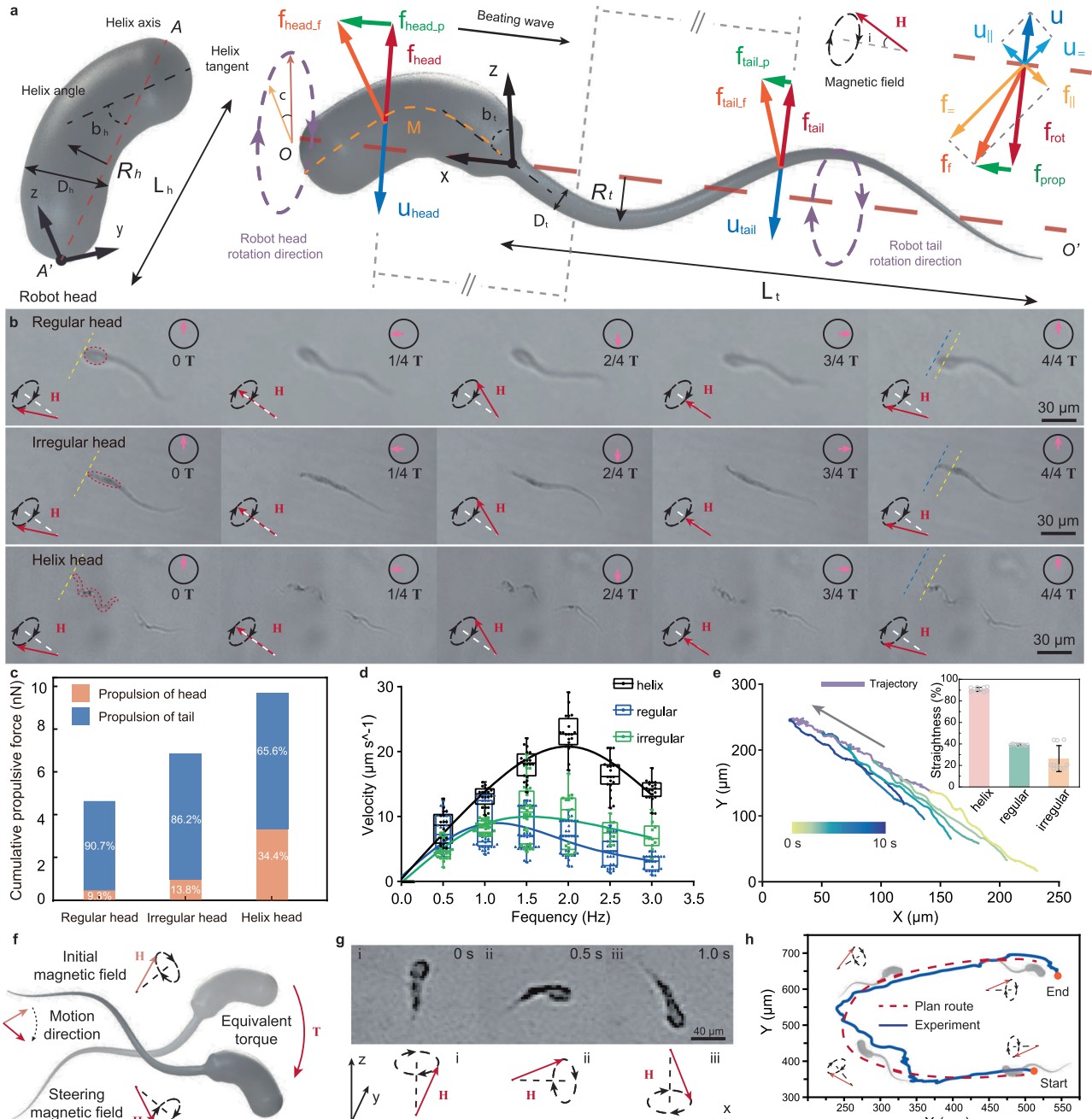

**Fig. 4 | Actuation and motion control of PSMs. a** The discretized geometry and mechanical analysis for dynamic modeling. The robot head is defined by diameter $D_h$, axis length $L_h$ and helix angle $\beta$, while the robot tail is defined by helix radius $R$, pitch $P$ and axis length $L_t$. Under the magnetic field, the rotation of both head and tail can provide propulsive force for locomotion. **b** The time-lapsed images show the different rotation states of microswimmer during locomotion. **c** The numerical calculations of propulsive force of different types microswimmers according to the proposed dynamic model. During which, the contribution ratio from robot head and tail are compared. **d** The velocity of diverse types microswimmers under the magnetic field with different frequency. Measured sample number is 10. **e** The comparison of locomotion straightness of three types microswimmers, and the locomotion trajectory indicates the robot with helix head performs better than the other two. Measured sample number is 10. **f** The control strategy of steering. **g** The bright field image of microswimmer's actual experimental steering. **h** The demonstration of controllable locomotion of microswimmer by moving along a preset C trajectory.

microswimmer is distributed along its long axis after fabrication. In theory, under the action of magnetic field, a magnetic torque **T** will be generated due to the misalignment angle $\gamma$ between the direction of the magnetic field **H** and equivalent easy magnetization axis **M**. Base on that, when a rotating magnetic field with a tilt angle $\theta$ about the locomotion direction OO' is applied, the head of microswimmer will revolve around locomotion direction OO' and rotate along its own helix axis AA' simultaneously (Supplementary Fig. 7). Following the revolving of head, the slender tail forms into a spiral shape (Supplementary Movie 2).

To elucidate the propulsion mechanism, we establish a general model for mechanical analysis of microswimmers with different shapes. From the view of geometry, the regular and irregular head of microswimmers can be considered as a specific helix whose helix angle is 90°. Therefore, all types of heads can be defined as rigid thick helix structures by helix angle $\beta_h$, helix length $L_h$, helix radius $R_h$ and

equivalent body width $D_h$. Similarly, the thin helix structure come from the flexible tail can be defined by original straightened length $S_t$, helix length $L_t$, helix radius $R_t$ and various body width $D_t$. The mechanical analysis indicates that the propulsion force of microswimmer comes from both head and tail. When considers the different body width between head and tail, we further proposed the dynamic model of microswimmer by combining the resistive force theory and the slender body theory (Supplementary Note 6). According to the obtained dynamic model, all of the dimensional characteristics of helix will affect the generated force and propulsion efficiency.

To intuitively show the differences of the microswimmers with different shape in propulsion, we record the DI water (pH-5.5–7.0) locomotion of three classical types of microswimmers (i.e., regular head, irregular head and helix head) in one gait cycle under the same magnetic field (strength **H** = 10 mT, tilt angle $\theta$ = 30° and frequency $f$ = 1 Hz) (Supplementary Movie 3). From the time-lapsed images in Fig. 4b and Supplementary Fig. 8, we can clearly observe the revolution and rotation of robot head, and the helical shape in slender tail. The results verify that our actuation and propulsion strategy is effective to all types of microswimmiers. During which, the microswimmer with helix head has higher locomotion efficiency than the other two, because the rotation of helix head along its own helix axis can provide additional propulsion force while that does not work in the axisymmetric regular and irregular heads. For instance, the microswimmer with helix head is able to move ahead -14.4 µm in one cycle, which is -1.8 times of the regular head (-8.2 µm per cycle) and -1.3 times of the irregular head (-10.7 µm per cycle).

To investigate the detailed effect of dimensional characteristics and the contribution of head/tail in propulsion, we quantitively calculate the propulsive force of polymorphous sperm-like microswimmers based on the proposed dynamic model. As the results shown in Fig. 4c and Supplementary Table S1, under the magnetic field with a strength of 10 mT, a frequency of 2 Hz and a tilt angle $\theta$ = 30°, the total propulsive force generated by the microswimmer with regular head, irregular head, and helix head is $46.1 \times 10^{-4}$ µN, $68.6 \times 10^{-4}$ µN and $97.3 \times 10^{-4}$ µN, respectively. During which, all types of robot head play a positive role in the propulsion. However, due to the different dimensional characteristics, the revolving and rotating of three types of head show different efficiency in breaking the symmetry of the deformation in time. For instance, the propulsive force from regular head, irregular head, and helix head are $4.3 \times 10^{-4}$ µN, $9.5 \times 10^{-4}$ µN and $33.5 \times 10^{-4}$ µN, which contributes to 9.3%, 13.8% and 34.4% to their total propulsive force respectively (Supplementary Note 7). Such higher efficiency of microswimmer with helix head consists well with the observed phenomenon in nature and the dual-helical propulsion mechanism[28].

Besides the theoretical analysis, we also conduct comparative experiments to evaluate the practical motion performance of microswimmer, including the velocity, motion straightness and steering controllability. We firstly measure the velocity of three types of microswimmers under 10 mT magnetic field with rotating frequency from 0.5 Hz to 3.0 Hz. As the results shown in Fig. 3d, the velocity of all microswimmers first rises and then falls as the frequency of the magnetic field increases. And the velocity of microswimmer with helix head is always larger than the other two, which increases from -9 µm s$^{-1}$ under 0.5 Hz to -22 µm s$^{-1}$ under 1.5 Hz then falls to -17 µm s$^{-1}$ under 3.0 Hz. In addition, we also analyze the average motion straightness of microswimmers with helix head, regular head and irregular head, which are 90%, 40%, and 30%, respectively (Fig. 4e). This is mainly caused by their difference in head swing amplitude and overall forward velocity, and the higher motion straightness means the better trajectory stability of robot. The steering of microswimmers is realized by adjusting the revolving direction of magnetic field, since an additional magnetic torque will generate and apply on the robot if its revolution axis is not coincident with that of rotating magnetic field (Fig. 4f). As

shown in Fig. 4g, although the existing of lagging response, the locomotion direction of microswimmer almost reverses as the direction of applied magnetic field turns 180°.

The proposed propulsion strategy enables the effective and controllable locomotion of constructed microswimmers, as evidenced by the controllable C motion trajectory (Fig. 4h), by emulating the sperm in nature, which provides the basis for potential applications of drug transportation and target delivery. Further, the polymorphous heads with different structure brings differentiated locomotion characteristics to the microswimmers, which could pave a path for the on-demand robot design and the differentiated control of multiple robots in the future.

## Gel material enabled pH-sensitive sustained release of microswimmers

Sustained release denotes a formulation intended to extend a drug's effectiveness by continuously releasing it within the body over an extended period. This process ensures a slow and non-constant release of the drug, contributing to its prolonged efficacy. The sustained release mechanism of the gel microswimmer is illustrated in Fig. 5a. Firstly, we stepwise coat chitosan (pH 6.5–6.6) and light concentration sodium alginate (0.03 w/v%) layers on the surface of the sperm-liked microswimmer (Fig. 5b and Supplementary Fig. 9, see Methods). Then, in order to obtain the ACA hollow microswimmer, the inner core of ACA coated microswimmer will be liquefied by destroying the substantial egg box structure and removing $Ca^{2+}$ from alginate inlayer under the ion exchange chelation sequentially (Methods and Supplementary Fig. 10 and Supplementary Movie 4). It's worth noting that such a reaction only occurs efficiently in weakly acidic to alkaline environment due to deprotonation[29,30] (Supplementary Fig. 11). As evidenced by the SEM images show in Fig. 5c, a thin ACA membrane (-5 µm) can be formed due to the tight bonding between the chitosan and alginate. Due to the penetration of the ACA membrane, the encapsulated molecules/drugs can be enhanced microswimmer stability, sustain released from the inner microswimmer into the external environment and selectively allowed the passage[15,31,32].

To reveal the drug release profiles of the sperm-liked polymorphous microswimmer, we investigate the effect of ACA membrane thickness on the cumulative release under the neutral environment (pH = 6.8–7.0). The experimental results show that the thickness can be well controlled by the reaction time due to their linear relation (Fig. 5d and Supplementary Fig. 12, and see Supplementary Movie 5), and the thickness of ACA membrane plays a major role in determining the release speed (Fig. 5e). We find that the sustain release can be achieved by choosing proper thickness of ACA membrane, as shown in Fig. 5e. If the thickness is too thick or thin, the release properties of the microswimmer cannot be maintained, because too thin or too thick thickness will respectively result in burst release due to not fully crosslinked between chitosan and sodium alginate amide bonds or the swelling reaction, and the microswimmers' swimming was hindered due to the increase in volume by swelling. (Supplementary Fig. 13 and Supplementary Note 8). However, the curve of release with a thickness around 6 µm is relatively flat -60%, and presents the sustainable release property. Therefore, the microswimmer is formed with a thickness of -6 µm, which will be selected for sustained release due to burst release effective remission under the neutral environment.

In the human body, the stomach typically maintains an acidic environment[33] (0.1 N HCl, pH-1.2), while the small intestine exhibits an alkaline environment[34] (phosphate buffer solution, pH ≥ 6.8). Therefore, we conducted the drug release test (the fluoresce dissolution tests under incremental pH values) in both acid and alkaline environments to demonstrate the potential of the microswimmers for potential drug release and evaluating the release stability. For comparison, we also take the uncoated ACA microswimmer as a control group. The results in Fig. 5f suggest that both microswimmers have

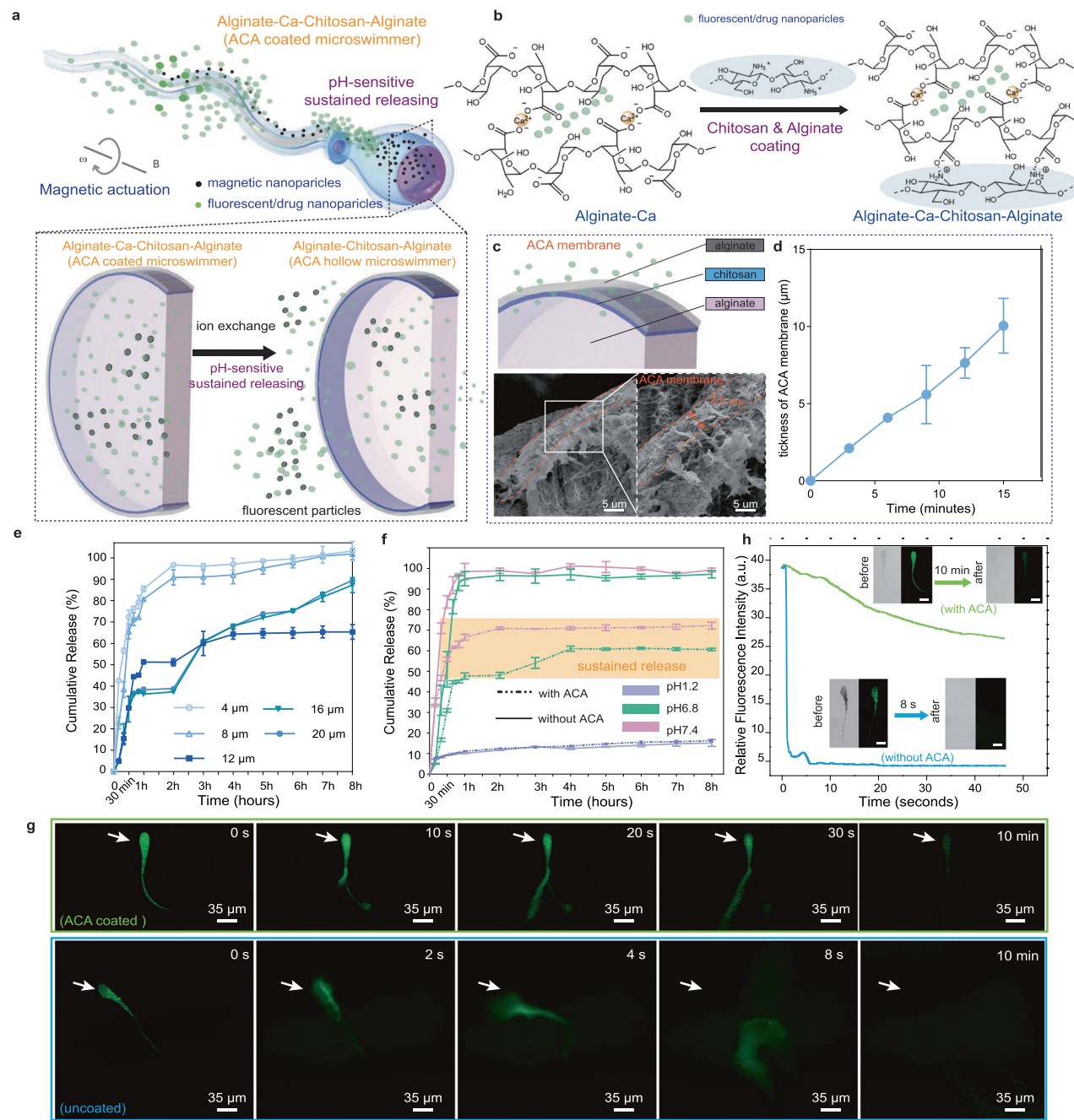

**Fig. 5 | pH-responsive sustained release of ACA-coated PSMs. a** Schematic representation of ACA-coated microswimmer sustained release drug/nanoparticles by ion exchange. **b** The formation mechanism of ACA membrane. Drug is wrapped by Alginate-Ca layers structure which is due to Na-alginate has ionic interaction and gelling abilities with divalent cations ($Ca^{2+}$), each $Ca^{2+}$ ion can bond with two carboxylic acid moieties of guluronic acid (G). Then, the positively charged amino groups ($NH_3^+$) of chitosan can with the negatively charged carboxyl groups ($COO^-$) of alginate and form a 3D cross-linking polyelectrolyte membrane. **c** The ACA membrane consists of three layers includes Alginate-Ca, Chitosan and Alginate. The SEM image shows ACA membrane when chitosan reaction during 9 min. **d** Relationship between the polyelectrolyte complexation reaction period and the ACA membrane thickness. The thickness of the ACA membrane increases with the increase of reaction time. Error bars indicate the standard deviation for $n = 7$ measurements at each data point. **e** The cumulative release of ACA coated microswimmers with different thickness ACA membrane. Error bars indicate the standard deviation for $n = 3$ measurements at each data point. **f** The release of FITC-dextran from the ACA coated microswimmers and uncoated microswimmer in different pH environment. Error bars indicate the standard deviation for $n = 3$ measurements at each data point. **g** Under in extreme alkali environment, the fluorescence image comparison between ACA coated microswimmers and uncoated microswimmer. **h** The curves of the fluorescence intensity between ACA coated microswimmers and uncoated microswimmer in extreme alkali environment. Scale bar is 20 μm.

superior stability in acid environment (pH = 1.2) because of their cumulative release is not exceed 12.6% for 8 h. However, with the increasing pH value until 6.8 or 7.4, the ACA-coated microswimmer can release 45% or 60%, respectively, after rapid increases before 30 min. On the contrary, the uncoated microswimmer within 30 min is as high as 88% and achieved full release in 1 h. Compared to that, the ACA

coated microswimmer until sustained the release of the drug between 30 min and 4 h, reaching 60% and 70% at 4 h and 2 h, respectively. These results suggest that ACA membrane is essential for the sustained release of the sperm-liked polymorphous microswimmer.

To demonstrate the sustained release capacity of ACA-coated microswimmer for drug delivery, we transfer the coated and uncoated

microswimmers in PBS solution, and then added sodium citrate (0.03 w/v%, pH=12 ± 0.5) to simulate the pH-responsive release transient process. As shown in Fig. 5g, ACA-coated microswimmer can maintain effective release for nearly 10 min in localized harsh pH environments (includes citrate, pH = 10–12) but an uncoated microswimmer only keep 8 s on sustained releasing. We also quantitively evaluate the release ability, the outer contour of microswimmer is selected as the area used to calculate fluorescence intensity for quantitative analyzing their sustained release capacities (Fig. 5h). The blue lines in the image have suddenly plummeted from 40 to 5 a.u. within 5 s, indicating that uncoated microswimmer appears burst release. Compared with the blue lines, the green line shows a slowly decreased tendency, demonstrating that ACA coated microswimmer has the capability to maintain its structure for retarding drug burst release. In addition, to further verify the microswimmers durability, the microswimmers is placed in a physiological saline solution (simulating the human electrolyte environment). The results show that the robot can maintain a relatively stable form within 45 min, providing the possibility for targeted drug delivery in the human body (Supplementary Fig. 14). These quantitative analyses prove the concept of using the PSMs for drug transportation and sustained release, which could be an alternative solution for targeted drug delivery in the future after further development.

## Discussion

Microswimmers owes both effective motion and cargo loading ability have potential in future biomedical applications. However, cumbersome manufacturing procedure, demanding requirements for applied materials, and polluted interruptions from external environments limit microswimmers' development. In this study, inspired by spermatozoa's morphology and function, we proposed an asymmetrical design with polymorphous head and flexible tail for endowing microswimmer both propelling and loading abilities at low Reynold environment. Different to the manufacture of traditional microswimmers[35–41] (Supplementary Table S2), the proposed VTAM approach enables the fabrication of biocompatible asymmetrical magnetic microswimmers with tail in one step. It not only provides high moveable ability but also achieves bionic semipermeable membrane encapsulation, offering sustain release capability for targeted drug delivery. The obtained PSMs can be precisely actuated and steered by rotating the magnetic field in low Reynold environment. Moreover, layer-packaged-based micoswimmer exhibits successful sustain release capability and release stability, which can be applied to an alternative solution for targeted drug delivery.

The manufactured polymorphous sperm-like magnetic microswimmers shed light on the design of micro swimming robots with excellent capability for motion efficiency, maneuverability, drug loading and sustained release, and will potentially benefit future biomedical treatment, especially the target delivery of diverse drugs and biologics. Using vortex turbulence-assisted microfluidics to fabricate sperm-like microswimmer with effective propelling, desirable loading and adaptive releasing abilities all in one, would inspire the following researchers to develop other sophisticated polymorphous structures fabrication that cannot be achieved by conventional laminar flow. It's worth mentioning that this manuscript mainly focuses on the proof of concept of the fabrication process. Compared to the conventional microcapsule fabrication approaches, there is a lot of room for improvement, such as fabrication controllability, stability, unity, and productivity. Moreover, further in vivo tests are also required for the structure optimization of the microswimmer to make it clinically applicable.

## Methods
### VTAM platform construction
Sodium alginate powder, anhydrous calcium chloride are purchased by m Acros Organics. Paraffin oil (light), Span 80 and $Fe_3O_4$ nanoparticle are obtained from Aladdin Co. Ltd (China). The microfluidic device is constructed by solidified PDMS bonded to glass slides by oxygen-plasma activation of both surfaces. Protruding microchannels are created via conventional soft lithography on silicon wafers with negative photoresist SU-8 50 (MicroChem). The photoresist is deposited onto clean silica wafers to a thickness of 40 μm for devices, a minimum of 25 μm or maximum of 50 μm for the width of channels, and patterned by ultraviolet light exposure through a photomask (CAD). PDMS prepolymer and curing agent are mixed in rations of 10:1 (w/w) and degassed in a vacuum pump. The molds are then baked at 70 °C for at least 2 h. After punching the inlet and outlet holes for fluid access, the chip is made a final combination with the surface of the glass slide. The two inlets of the microfluidic channels are connected by polyethylene tubing to plastic syringes installed in syringe pumps. The glass capillary with an inner diameter 0.1 mm is used to connect the outlet to 0.5–2.5 wt% $CaCl_2$ solution stirred by a magnetic stirrer.

### Fabrication of monodispersed magnetic alginate/oil droplet
The magnetic alginate/oil droplet is generated using a cross-shaped microfluidic device. Briefly, the paraffin oil with 0.1 w/v% Span 80 is injected into the microfluidic chip through vertical channels as a continuous phase. The 1 wt% sodium alginate mixes with 1:1 volume ratio $Fe_3O_4$ nanoparticle is injected into horizontal channels as dispersion phase. When the continuous and dispersion phase flow rates are 1.5 and 0.5 μl min$^{-1}$, the alginate-$Fe_3O_4$ solution is sheared and formed into a monodisperse droplet by oil phase in cross-shaped channels. The iron oxide particles and drug are distributed in the whole body (Supplementary Fig. 15).

### Measurement of the density and viscosity
To measure the density, we filled the solution in a 1 ml PE tube, and then measure the solution mass (*m*) on a scale (RADWAG AS 220.R2 PLUS, Poland). The density of solutions was calculated by dividing the mass by volume (*m/v*). About viscosity—The viscosity was measured by viscometers (Techcom SNB-2E) at 25 °C in a thermostatic circulating water bath (DC-0506).

### Effect of the Ca²⁺ concentration
The 1 wt% cap-type gels are produced through an external gelation method (as depicted in Supplementary Fig. 6), wherein 3 g sodium alginate solution is introduced into a porous plastic cap mold. By immersing these caps into various concentrations of $CaCl_2$ solution (ranging from 0.5–2.5 wt%), we observed distinctive gelling behaviors.

### Electromagnetic coils set up
The magnetic control system comprises three sets of coils, each with distinct sizes (Supplementary Fig. 16). Specifically, the coil radii are 20 cm, 14 cm, and 8 cm, respectively. Each pair of coils consists of two identical coils positioned at a separation distance equal to their individual radii. The Helmholtz coil is characterized by extremely uniform magnetic field space, and its magnetic field strength has a favorable linear relationship with the supply current that encourages the manufacture of the three-dimensional combined magnetic field. The normal operation of the entire system is composed of six sides electromagnetic coils, a computer, a data acquisition board, a camera and six drivers. The system can generate arbitrary 3D magnetic fields up to 10 mT in magnitude and can operate at a maximum frequency of 50 Hz. The settings and adjustments of parameters such as the rotation direction, magnetic field strength, rotation frequency are performed through a computer client. The data acquisition board connected to the computer can convert control commands into voltage signals for transmission to the driver. Then, the driver outputs a corresponding current based on the received signal to the electromagnetic coil to generate magnetism. The USB microscope can obtain the robot

motion video and send the image to the computer for viewing by the operator.

## Average motion straightness calculation

The term average motion straightness refers to the calculation of motion straightness, which is defined as the ratio of VSL (straight-line velocity) to VAP (average path velocity). To derive the average motion straightness, we calculate this ratio for each robot's trajectory and then compute the average of these parameters. Specifically: motion straightness = VSL/VAP, where: VSL represents the straight-line velocity, VAP denotes the average path of the robot divided by the total tracking time. This metric of motion straightness helps in characterizing the overall motion patterns of the robots under study.

## Formation of ACA hollow microswimmer

The calcified microswimmer in the centrifuged tube is deoiled by washing with DI water. Then, the chitosan solution with 0.1–0.5% (w/v) is added into the calcified microswimmer in a centrifuged tube. Gently aspirating with a pipette repeatedly has ensured that the calcified microswimmer is enwrapped evenly by chitosan for 3–15 min. Next, the tube is centrifuged at $500 \times g$ for 1 min and then the solution is removed. After centrifugation, DI water is used to remove excess chitosan solution that has not reacted with sodium alginate. These methods are aimed at minimizing aggregation and preserving the individual integrity of the microswimmers throughout the experimental processes. Meanwhile, the chitosan-coated microswimmer is washed and transferred into another tube filled with 0.03% (w/v) sodium alginate solution and the alginate coating as well as the chitosan coating process are carried out. Eventually, the ACA coated microswimmer is formed after repeated cleaning of excess alginate with DI water. These precautions are crucial in ensuring accurate experimental outcomes and maintaining the microswimmers' integrity.

## In vitro ACA-coated microswimmer drug dissolution test

To determine the drug release of the ACA-coated microswimmer, 1 wt% sodium alginate containing about 0.05% (v/v) fluorescent nanoparticles solution is injected as the inner phase to form the ACA-coated microswimmer. The ACA-coated microswimmer is equilibrated to room temperate and washed with a copious amount of water for 10 min before the drug release measurements. Independent FITC-dextran-loaded microswimmer arrays are stimulated with HCl (0.1 N, pH 1.2) and PBS buffer (0.1 mol L$^{-1}$, pH 6.8 and 7.4) solutions to assess the release. Fluorescence intensities (bandpass emission: 447 nm) over six microswimmers in each group are measured using the Nikon software. Background fluorescence is subtracted from the measured values.

## Data availability

The data used for showing locomotion straightness of three types microswimmers at https://doi.org/10.6084/m9.figshare.25816231.v2, whereas the velocity data used in micrswimmer locomotion is available at https://doi.org/10.6084/m9.figshare.25816237.v2. The data used for showing ACA membrane thickness at https://doi.org/10.6084/m9.figshare.25816240.v2, whereas the cumulative release of ACA-coated microswimmers data is available at https://doi.org/10.6084/m9.figshare.25816243.v2 and https://doi.org/10.6084/m9.figshare.25816234.v2. Other data needed to evaluate the conclusions are provided in the main text, Supplementary files with this paper.

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

## Acknowledgements
This work was supported by the National Science Foundation of China (NSFC)/Research Grants Council (RGC) Joint Research Scheme (JRS) (N_HKUST638/23), the Hong Kong RGC General Research Fund (11216421) and Shenzhen-Hong Kong-Macau Science and Technology Project (Category C) SGDX20201103093003017.

## Author contributions
R.T. and X.Y. did the experiment and analyzed the data. H.L. helped in polishing the manuscript. Y.S. initialized the idea and designed the experiment. R.T. and X.Y. contributed equally to this work.

## Competing interests
The authors declare no competing interests.
