## [Peer Review File · Nature Communications]

Reviewers' comments:

Reviewer #1 (Remarks to the Author):

In this paper, the authors reported a novel strategy to fabricate the sperm-like swimming microrobots using vortex turbulence. Utilizing microfluidic technique, the microswimmers consisting of alginate, chitosan, and iron oxide particles, were developed in one step. The resulting microswimmers are fully biocompatible and degradable, which resolve the issue toward biomedical applications. Particularly the microswimmers can load and release drug in a controllable manner. The proposed idea is very promising and the manuscript is clearly written to demonstrate the effect of the sperm-like microswimmers. Below are some technical comments for the authors' consideration to further improve the manuscript prior to its publication.

1. In case of the fabrication design, this reviewer concerns the quality of the microswimmers. Can you control the geometry of the swimmer, such as the length, thickness, pitch, flexibility, and mechanical strength? As the alginate-based microswimmers were crosslinked by calcium ions, what is durability of the microswimmers in long term, particularly in solution with sodium ion, such as saline? What is the distribution of iron oxide particles in the microswimmers, in whole body or just head of swimmers? Moreover, were the fabricated microswimmers liable to aggregate after the deoiling process, and centrifugation steps during washing steps and coating of chitosan and alginate in ACA? Considered the work mainly focus on the fabrication of microswimmers, this reviewer recommends the authors to show the data for mass production of the microswimmers.

2. For the magnetic actuation, the author compared the movement behavior of swimmers with helical head, regular head, and irregular head. Considered that the propulsion mechanism as natural sperm, which is mainly attributed to the deformation of their body and generated deformative wave. So this reviewer feel that the geometrical parameters of swimmers including length, flexibility, diameter (or thickness, not pitch) of swimmers, which can be controlled by exerting the fabrication method, may be more important to investigate.

3. For drug release, the authors design an AOA coating-based controllable release strategy, the resulting microswimmers enable the release of model drug in alkaline pH environment. This reviewer is not an expert in drug release, and he is not so understand what case would be utilized the drug release in alkaline pH environment. It may be better that the author discuss the application situation of microswimmer-based pH-sensitive drug release. The reviewer gently suggests the drug release of microswimmers may be important enough, maybe the author do not have to concern the drug release in alkaline pH environment.

Reviewer #2 (Remarks to the Author):

The manuscript presents the fabrication and testing of sperm-like micro-swimmers.

The fabrication process is based on the microfluidic formation of monodisperse water-in-oil emulsions, followed by crosslinking of the water phase into a hydrogel under vigorous stirring.

Three different micro-swimmer morphologies are obtained by tuning the process parameters.

The hydrogel microstructures are loaded with iron oxide particles and actuated by precessing magnetic fields, resulting in (open-loop)controlled swimming.

Finally, the micro-swimmers are modified with a chitosan-alginate coating and their ability to release a model molecule is evaluated.

The fabrication and swimming of the microstructures are supported by theoretical analysis.

The manuscript is well organised and presented (although extensive proofreading is needed).

The reported results and methods have a moderate degree of novelty and of relevance to the reference community.

The soundness of the reported research needs improvements, as detailed in the following.

- Substantial and extensive modelling of different phenomena is reported in the manuscript (Supplementary Information), ranging from the droplet formation to the swimming of the fabricated microstructures and their swelling behaviour. Nonetheless, there is no direct comparison of theoretical and experimental results throughout the manuscript. In presenting and discussing your results, please compare them with theoretical predictions, to both provide an understanding of the experimental observations and validate the presented theory.

- How much is the diameter of the capillary realising the droplets in the vortex?

- Is λ the alginate-water or alginate-oil interface thickness? The physical meaning and the role of these lengths are not clear: please provide a clearer explanation.

- The dependence of the tail length on the flow speed and on the Ca^{2+} concentration is mentioned, but no evidence is reported: please provide such evidence.

- The numerical simulations of the demulsification process are performed in a geometry that is very different from that adopted experimentally. The results are thus probably not so relevant to the experimental observations.

- How are the simulation parameters set? Were the viscosities measured? How can the viscosity of CaCl₂ solutions change so much at such low concentrations? I expect the viscosity of the solution to change very little with respect to the concentration of CaCl₂. Therefore I believe that the viscosity cannot be used as a regulation parameter. Please comment on this.
- Which variable do the colours in Suppl. Fig. 6 represent?
- In which liquid was the magnetic swimming tested?
- A detailed description of the magnetic field generator is missing: please provide more details, together with a picture and/or a schematic. If the three orthogonal pairs of coils are the same, they cannot be Helmholtz coils: in Helmholtz coils, the distance between the two coils equals the radius of the coils.
- There are several mentions of a "beating wave": as they undergo head precession and tail rotation, beating waves are likely not generated at all. Swimming is likely achieved by helical propulsion.
- Page 8: 14.4 mm should read 14.4 microns; by "circle" do you mean "cycle"?
- What do you mean by "average motion straightness"? How is it calculated?
- Please clarify how the alginate-chitosan bilayer shell forms. Also, why are three layers needed?
- Can you provide any evidence of liquefaction of the alginate core in alkaline environments? And which is the reason for liquefying the core?
- "Sustainable" and "sustain" release are mentioned: did you mean "sustained release"? This is not clearly characterized (the fact the release curve flatten does not imply that all drug molecules will be released at a controlled rate).
- Are magnetic nanoparticles loaded into the realising microswimmers? Please explain this in more detail.
- What do the low and high pH represent from a physiological point of view? In which applications can you encounter these extremes? Also, there seems to be very little difference between pH = 7.4 (physiological) and 6.5 (tumour micro-environment, inflamed tissues).
- What is the swelling behaviour reported in Suppl. Fig. 10? Please provide details on these experiments.
- A proper comparison of the proposed process and micro-swimmers with those used for developing other sperm-like microrobots should be provided.

Reviewer #3 (Remarks to the Author):

The main novelty of this manuscript is that the authors report a method to fabricate alginate structures with elongated "tails" that, when rotated by a magnetic field, can be propelled. The method requires only one-step in a microfluidic device attached to a vortex flow. The fabricated structures can take advantage of the controllable release of substances available to alginate microspheres.

In my opinion this manuscript does not rise to the level of significance to be published in Nature Communications. The authors report the new fabrication method, but that is about all. There is no new knowledge gained by using the new devices in an experiment, or new application areas that the devices are used for. This is detailed below.

The authors do provide some simulations to try to provide "physical principles" of their fabrication method. However, I did not see much explanation or understanding gained from these simulations. Mostly, the meat of what was actually applied was an analysis of the relevant dimensionless parameters, then a variation of them (by the vortex flow rotation speed and viscosity) to show in what parameter ranges they find different fabricated morphologies. The simulations were not actually necessary for this type of parameter-space exploration. This type of parameter space exploration doesn't really provide any physical insight either.

The simulations themselves (described mostly in SM) were not very illuminating. As the authors say in p8 of the SM, their results for the helix head were not observed in the model. Also, some of the writing was unclear and not many results were presented of the simulations except for a couple examples. For example, in the sentence after Eq S23, is that a statement about the simulation results, or a statement about the experiments? It would help if maybe the authors were referring to some specific (experimental or simulation) results, but there don't appear to be anything supporting that statement in the simulation results presented.

The method doesn't seem really controllable to make other shapes and applications possible, at least based on what the authors have said. They don't provide an examples of these, for instance. This is made worse since there is not much physical insight into the fabrication method.

The numerical calculations of swimming are standard but not really well-described. For example, how is the helical shape of the flexible tail under actuation determined? Is it visualized during rotation? Or do the authors try a large number of possibilities? Since due to flexibility the shape is likely different under rotation than when still, it is not clear how these are accurately estimated.

minor:

-In Introduction 2nd paragraph, 1st sentence. The cell membrane is not a thin rim of cytoplasm.

-3rd and 4th lines of p3: microspheres are suitable for propulsion near boundaries to fluids, as has been demonstrated in an extensive body of literature. They are also suitable for propulsion in non-Newtonian fluids.

Response to reviews for the manuscript

We would like to thank all reviewers for thoroughly reading our manuscript entitled “**One-step formation of polymorphous sperm-like microswimmers by vortex turbulence assisted microfluidics**”. Those comments are helpful for raising the quality of the paper, as well as with the important guiding significance to our research.

Regarding the main concern on the processing mechanism and simulation results, in this revised version, we have (1) added more experiments (microswimmer production under different calcium ion concentrations and magnetic stirrer rotation speed), (2) elucidated the fabrication process (encapsulated droplet production, slender tail extraction, and sperm-liked morphology solidification), (3) optimized the simulation model (investigation of the effect of the shear velocity on the generated structure of the microswimmers), and (4) conducted additional data analysis (detailed comparison of our experimental observations alongside theoretical and simulation predictions). Moreover, we have also checked for spelling and grammar errors and adjusted relevant descriptions. We believe that the questions raised by the reviewers have been well addressed in this revised version.

In this response letter, we answered the comments in a point-by-point manner. For the convenience of the reviewers, the comments and suggestions are listed below in the *blue font*, followed by our responses in the normal black font. We also highlighted the corresponding modification in our revised manuscript in the *red font*.

Reviewer #1:

In this paper, the authors reported a novel strategy to fabricate the sperm-like swimming microrobots using vortex turbulence. Utilizing microfluidic technique, the microswimmers consisting of alginate, chitosan, and iron oxide particles, were developed in one step. The resulting microswimmers are fully biocompatible and degradable, which resolve the issue toward biomedical applications. Particularly the microswimmers can load and release drug in a controllable manner. The proposed idea is very promising and the manuscript is clearly written to demonstrate the effect of the sperm-like microswimmers. Below are some technical comments for the authors' consideration to further improve the manuscript prior to its publication.

Response: Thanks for your meticulous evaluation of our work. In response to your suggestions on microswimmer production and control, we have conducted additional experiments to substantiate the correlation between the tail design and its influence on flow speed. Furthermore, we have provided conclusive evidence supporting the production of polymorphous sperm-like microswimmers (PSMs). Please find the detailed responses below.

1. In case of the fabrication design, this reviewer concerns the quality of the microswimmers. Can you control the geometry of the swimmer, such as the length, thickness, pitch, flexibility, and mechanical strength?

Response: Thanks for your comments. The length and thickness of the swimmer can be controlled by the rotational speed and chitosan coating time. The flexibility and mechanical strength of the swimmer can be adjusted by the gelling time and CaCl_2 concentration. As to the pitch, it's related to various parameters like flow speed, gelling time, etc. Because this manuscript mainly focuses on the fabrication of sperm-like tails, we'd like to study pitch control in our future work. Please find the detailed explanation below:

(1) Length: Figure R1 illustrates the growth of the regular microswimmer's tail with increasing rotation speed (800-1000 rpm), showcasing the length control by rotational speed. Details about the results and explanation are given in new section 'Effect of flow speed/ Ca^{2+} concentration on polymorphous tailed-magnetic microswimmers formation' and Maintext Figure 3.

Figure R1. As the rotational speed increases from 800 to 1000 rpm, the regular microswimmer's body undergoes gradual elongation and exhibits the emergence of a tail-like structure.

(2) Thickness: The experimental results verify that the thickness is proportional to the continuous chitosan coating time (Supplementary video 5, Figure R2). Similar results can also be found in Figure 4D, where ACA membrane thickness thickens with chitosan encapsulation time.

Figure R2. The thickness of the spherical microrobot increases with the duration of chitosan coating.

(3) Strength and flexibility: The flexibility and mechanical strength of the microswimmer are determined by gelling time and CaCl_2 concentration. The increase of gelling time and CaCl_2 concentration will lead to an increase in the gel crosslinking density¹⁻⁴. Gel strength increases with gelation time as long as sufficient unbound calcium and alginate are present. Thus, the flexibility and mechanical strength of microswimmers can be enhanced by elevating the CaCl_2 concentration or accelerating the gel speed.

We have added these results and discussion in the “Gel material enabled pH-sensitive sustained release of microswimmers” section (Page 12 line 29), Supplementary Figure S11 and Supplementary video 5.

2. As the alginate-based microswimmers were crosslinked by calcium ions, what is durability of the microswimmers in long term, particularly in solution with sodium ion, such as saline?

Response: Based on your comments, we have evaluated the durability of the microswimmer in 0.9% sodium chloride saline solution in this revised version (Figure R3). The results suggest that it can maintain stable for 45 minutes, and preserve for 75 minutes before eventual rupture. This durability time is sufficient for robot actuation and drug release.

We have provided related descriptions and results in the ‘Gel material enabled pH-sensitive sustained release of microswimmers’ section (Page 13, line 9) and Supplementary Figure S13.

Figure R3. The durability of alginate-based microswimmers in 0.9% sodium chloride saline solution (pH~5.5).

3. *What is the distribution of iron oxide particles in the microswimmers, in whole body or just head of swimmers?*

Response: The iron oxide particles are distributed in the whole body, as the microscope image shows in Figure R4. The reason is that iron oxide particles are incorporated into the initial sodium alginate solution, ensuring their presence across the entire body of the microswimmers rather than being concentrated solely in specific regions.

We have provided related results and explanations in Method Section ‘Fabrication of monodispersed magnetic alginate/oil droplet’ (Page 15, line 1) and Supplementary Figure S14.

Figure R4. The distribution of iron oxide particles in the microswimmers.

4. *Moreover, were the fabricated microswimmers liable to aggregate after the deoiling process, and centrifugation steps during washing steps and coating of chitosan and alginate in ACA?*

Response: The aggregation in ACA is a potential issue in fabricating the microswimmer. To address this challenge, we have employed gentle pipetting techniques and utilized DI water to wash excess uncross-linked solution and prevent its aggregation with another microswimmer’s uncross-linked layer after each coating step (Figure R5). The results suggest that these measures can minimize aggregation and preserve the individual integrity of the microswimmers throughout the experimental processes. We have provided the description of the fabrication process in the Method Section ‘Formation of ACA coated microswimmer’ (Page 15, lines 6 and 11) and Supplementary Figure S9.

Figure R5. The process of ACA coating on the surface of PSMs.

5. Considered the work mainly focus on the fabrication of microswimmers, this reviewer recommends the authors to show the data for mass production of the microswimmers.

Response: Thank you for your valuable suggestion. The data for the mass production of the microswimmers is shown in Figure R6. Briefly, a notably high production yield of helix head microswimmers at an optimal rotation speed of 800 rpm and a 1.5% calcium chloride solution concentration. The regular microswimmers will be massively produced by elevating the rotational speed to 1000 rpm and adjusting the calcium ion concentration to 2%, as evident in Figure R6(iv).

We have provided a related description and results in ‘The formation of polymorphous tailed-magnetic microswimmers’ section (Page 5 line 1) and Maintext Figure 2.

Figure R6. The production of polymorphous sperm-like microswimmers (PSMs).

6. For the magnetic actuation, the author compared the movement behavior of swimmers with helical head, regular head, and irregular head. Considered that the propulsion mechanism as natural sperm, which is mainly attributed to the deformation of their body and generated deformative wave. So this reviewer feel that the geometrical parameters of swimmers including length, flexibility, diameter (or thickness, not pitch) of swimmers, which can be controlled by exerting the fabrication method, may be more important to investigate.

Response: We agree with your comments that the geometrical parameters are important for the motion of the microswimmer. This is the major reason why we develop the sperm-like structure with tails. As the results shown in Fig. R7, we have revealed that the microswimmers with various head shapes have distinct propulsion contributions under identical actuation conditions, i.e., the helical head yields the most substantial proportion of propulsion force (34.4%), followed by the irregular head (13.8%), and then the regular head with the smallest contribution (9.3%). Notably, we find that there is a significant disparity in the propulsion force generated by different heads, ranging from 2 to 8 times, while the variance in propulsion forces from different tails is comparatively smaller, ranging from 1.5 to 2 times. This discrepancy is attributed to the inherent characteristics of the head and tail, with the head being rigid and the tail being soft, as detailed in the main

text page 9-10 and Supplementary Section 6. Therefore, in this manuscript, we have primarily focused on investigating the impact of head structure, i.e., regular, irregular, helix, on propulsion. In the future work, we will broaden our scope to investigate the influence of additional parameters on microswimmer motion and performance, like the materials, size, etc as you suggested.

We have provided related discussions in “Actuation and motion control of microswimmers” section (main text Page 9-10).

Figure R7. The production of polymorphous sperm-like microswimmers (PSMs).

7. For drug release, the authors design an AOA coating-based controllable release strategy, the resulting microswimmers enable the release of model drug in alkaline pH environment. This reviewer is not an expert in drug release, and he is not so understand what case would be utilized the drug release in alkaline pH environment. It may be better that the author discuss the application situation of microswimmer-based pH-sensitive drug release. The reviewer gently suggests the drug release of microswimmers may be important enough, maybe the author do not have to concern the drug release in alkaline pH environment.

Response: In the human body, the stomach typically maintains an acidic environment (pH~1.2), while the small intestine exhibits an alkaline environment (pH≥6.8). Therefore, we have conducted the drug release test in both acid and alkaline environments to demonstrate the potential of the microswimmers for potential drug release.

Fig. 4f demonstrates the stability tests that simulate diverse pH environments similar to those within the human body. The results indicate that the microswimmers can remain stable in acidic environments, while achieving sustained drug release in neutral and alkaline environments. Upon carefully analysis, these outcomes signify the potential biomedical relevance of our microswimmers in drug delivery. For instance, the microswimmers can be designed to withstand the acidic stomach environment, and be delivered through the digestive tract, then release their cargo upon entering the alkaline environment of the small intestine.

We have provided related description and results in ‘Gel material enabled pH-sensitive sustained release of microswimmers section’ (Page 12 line 14).

Reviewer #2:

The manuscript presents the fabrication and testing of sperm-like micro-swimmers. The fabrication process is based on the microfluidic formation of monodisperse water-in-oil emulsions, followed by crosslinking of the water phase into a hydrogel under vigorous stirring. Three different micro-swimmer morphologies are obtained by tuning the process parameters. The hydrogel microstructures are loaded with iron oxide particles and actuated by precessing magnetic fields, resulting in (open-loop) controlled swimming. Finally, the micro-swimmers are modified with a chitosan-alginate coating and their ability to release a model molecule is evaluated. The fabrication and swimming of the microstructures are supported by theoretical analysis. The manuscript is well organised and presented (although extensive proofreading is needed). The reported results and methods have a moderate degree of novelty and of relevance to the reference community. The soundness of the reported research needs improvements, as detailed in the following. Substantial and extensive modelling of different phenomena is reported in the manuscript (Supplementary Information), ranging from the droplet formation to the swimming of the fabricated microstructures and their swelling behaviour. Nonetheless, there is no direct comparison of theoretical and experimental results throughout the manuscript. In presenting and discussing your results, please compare them with theoretical predictions, to both provide an understanding of the experimental observations and validate the presented theory.

Response: We sincerely thank you for your detailed comments on our manuscript. The acknowledgment of our work's organization and presentation is deeply appreciated. Your constructive feedback has been invaluable in improving the clarity and accuracy of our content, prompting us to proofread work to ensure maximum coherence and accuracy in our manuscript.

In response to your observations regarding the comparison between theoretical predictions and experimental results, we have undertaken a comparative analysis between our experimental observations and theoretical predictions throughout the manuscript. (Maintext Figure 3, Supplementary Section 5). Furthermore, we have made significant progress in extending the working depth by providing detailed simulation parameters and outlining the magnetic field settings in this revised rendition (Method Section 'Electromagnetic coils set up', Supplementary Section 3 and Supplementary Figure S15). These additions are aimed at providing a more comprehensive framework for illustrating the phenomena we report. Please check the detail response below.

1. How much is the diameter of the capillary realising the droplets in the vortex?

Response: The diameter of the glass capillary is 100 μm in our VTAM platform. For detailed information, please refer to the Method Section 'VTAM platform construction' (Page 14, line 21).

2. Is lambda the alginate-water or alginate-oil interface thickness? The physical meaning and the role of these lengths are not clear: please provide a clearer explanation.

Response: Thank you for your meticulous review and for bringing attention to this aspect of our study. “ λ ” specifically denotes the thickness of the alginate-oil interface.

The thickness of the alginate oil interface (λ) is determined using a Ginzburg-Landau free energy function-based thermodynamics formula: $\lambda = 4b\kappa/3\sigma$, where 'b' represents a constant coefficient, ' κ ' corresponds to surface curvature, and ' σ ' signifies interfacial tension. This parameter plays a pivotal role in the demulsification process during fabrication. Our analysis reveals that λ should be designed negligible and small enough to ensure the demulsification in the generated vortex.

We have provided a more detailed and clarified explanation in the revised version, particularly on the Maintext Page 5 Line 23 and Supplementary Information Page 4 Equation 6 and 7, elaborating further on this crucial aspect.

3. The dependence of the tail length on the flow speed and on the Ca^{2+} concentration is mentioned, but no evidence is reported: please provide such evidence

Response: We'd like to clarify that the tail length is mainly determined by the flow speed. The Ca^{2+} concentration isn't directly relevant to the tail length but plays a major role in the gelling of the microswimmer. Please find the detailed explanation and evidence as below:

Effect of the flow speed: The theoretical model (Maintext Page 7 Line 13 and Supplementary Section 5) implies that the tail of the microswimmer can be formed when the shear velocity falls within the range of 0.6 m/s to 1 m/s (corresponding to speeds of 800-1000 rpm). In this revised version, we have verified this conclusion by conducting a series of experiments. As illustrated in Fig. R8, when the rotation speed changed from 800rpm to 1000 rpm, we can observe the increase in the length of the microswimmers' tail. This statement is also validated through simulation, as the results shown in Fig. R8, affirming the influence of shear velocity on the fabrication process of microswimmers.

Figure R8. The comparison of regular head microswimmer's morphology between experimental and simulated conditions at rotational speeds ranging from 800 to 100 rpm.

Effect of the Ca^{2+} concentration: As the results shown in Figure. R9, the concentration of Ca^{2+} plays a pivotal role in the solidification process. Specifically, when the concentration of Ca^{2+} falls below 1 wt% or surpasses 2.5 wt%, the tail-shaped microswimmer cannot be formed. Within the concentration range of 1 wt% to 2.5 wt%, the Ca^{2+} concentration doesn't exert a significant influence on the length of the tail. Within this optimal Ca^{2+} concentration range (1%-2.5%), the calcium ion concentration proves adequate for the solidification of the elongated microswimmer, and the primary factor affecting the tail length is the flow speed.

In summary, in this revised version, we have confirmed that the direct impact of flow velocity on tail length, and the influence of Ca^{2+} concentration on the gelling speed of microswimmers. Together, these factors play a crucial role in shaping the morphology of the microswimmers. In this revised version, we have incorporated these new data and delved into the discussion on the effect of shear velocity on tail length. For further details, please refer to Maintext Page 8 Line 2 and Supplementary Section 5 in the manuscript.

Figure R9. The morphology of microswimmers generated by the VTAM platform under different calcium ion concentrations.

4. The numerical simulations of the demulsification process are performed in a geometry that is very different from that adopted experimentally. The results are thus probably not so relevant to the experimental observations.

Response: The purpose of the simulation is to explore the impact of the shear velocity v_{ve} on the formation of the microswimmers. We choose a 2D environment to conduct the simulation based on two primary considerations: (1) Firstly, the critical parameters for demulsification do not heavily rely on the microswimmer's 3D geometry, and simplifying the simulation in a 2D environment is a reasonable choice. (2) Secondly, conducting the simulation in a 2D environment offers high exploration efficiency and computational resource-saving. This approach aligns with established practices in fluidic field research, where 2D simulations have been commonly employed for analyzing deformation and velocity properties in 3D flow⁵⁻⁷. Hence, we have selected the 2D plane as the simulation environment for this manuscript.

As the results shown in maintext Fig. 2, the simulation in 2D can successfully capture the shaping on the oil-water interface and the subsequent breakage of the oil film under the impact of vortex flow. These outcomes align well with both theoretical analyses and experimental results, verifying the result obtained from the simulation is valuable. Guided by these simulations, we can predict the demulsification success rate in the experiment, which is very useful for experimental setup.

We thank your insightful comments on the simulation, and will try to build up a 3D model to conduct a more precise analysis in our future work for more complex structure formation.

5. How are the simulation parameters set? Were the viscosities measured? How can the viscosity of CaCl₂ solutions change so much at such low concentrations? I expect the viscosity of the solution to change very little with respect to the concentration of CaCl₂. Therefore I believe that the viscosity cannot be used as a regulation parameter. Please comment on this.

Response: In the simulation, the density of the alginate, oil, and CaCl₂ solution is set as 1.6×10^4 kg/m³, 800 kg/m³ and 1000~1025 kg/m³, respectively. The dynamic viscosity of the alginate, oil, and CaCl₂ solution is set as 0.0137 Pa.s, 0.032 Pa.s, and 0.00325 Pa.s, respectively. For the geometric parameters, the droplet diameter, oil thickness, and inner diameter of the glass tube is set as 50 μ m, 0.5 μ m, and 100 μ m, respectively. These parameters are set based on the measurements in the experiments.

We agree with you that the Ca²⁺ concentration doesn't affect the viscosity so much. The Ca²⁺ concentration mainly affects the gelling process, and then determines microswimmer's formation. We regret that we didn't explain this clearly in our previous manuscript. In this revised version, we have rewritten the relevant parts to explain the formation process of the microswimmers and the effect of Ca²⁺ concentration on the solidification. Please find the contents in Maintext Page 8 Line 2 and Supplementary Section 5 for details.

We have conducted experiments to validate the effect of Ca^{2+} concentration on gelling speed. As depicted in Figure. R10A, (i) at < 1 wt% CaCl_2 concentration, no cap gels formed after 60 minutes due to insufficient calcium available to bind carboxyl-sites of alginate. The results align with prior studies indicating that lower concentrations of calcium solution hinder gel formation (unable to form a tail-shaped microswimmer, as seen in Figure. R11i). (ii) When the Ca^{2+} concentration is high (>2.5 wt%), the microswimmer gels immediately before it is impacted and extracted by the flow. This rapid solidification, induced by high Ca^{2+} concentrations upon contact with sodium alginate droplets, results in round or nearly round microswimmers (as seen in Figure. R11iv). (iii) Within the range of 1-2 wt% CaCl_2 , there almost no difference in the gel morphology. These comparison results indicate that the concentration of calcium ions has minimal impact on the formation process.

In short, the Ca^{2+} concentrations within the range of 1-2 wt% yield a higher number of microswimmers at consistent rotation speeds.

Figure R10. Representative alginate gel images showing the pattern of alginate gel formation in a cap-shaped mold using external gelation method as a function of time (15–60 min) and CaCl_2 concentrations (0.5–2.5 wt%). Na-alginate solution concentration is 1 wt %.

Figure R11. The morphology of microswimmers generated by the VTAM platform under different calcium ion concentrations.

6. Which variable do the colours in Suppl. Fig. 6 represent?

Response: The colors depicted in Supplementary Figure 6 represent the surface concentration of alginate droplets, measured in mol/m^3 . We have taken note of this query and have revised Maintext Figure 3a accordingly.

7. In which liquid was the magnetic swimming tested?

Response: The magnetic swimming capabilities of our microswimmers are evaluated in a controlled environment using deionized (DI) water. The choice of DI water, devoid of ions to ensure stability (pH~5.5), is deliberate due to the sensitivity of our microswimmers to pH variations, which could potentially induce structural damage via ion displacement reactions. We have detailed the rationale for selecting DI water on Maintext ‘Actuation and motion control of microswimmers’ (Page 9 line 21) for further clarification.

8. A detailed description of the magnetic field generator is missing: please provide more details, together with a picture and/or a schematic. If the three orthogonal pairs of coils are the same, they cannot be Helmholtz coils: in Helmholtz coils, the distance between the two coils equals the radius of the coils.

Response: The magnetic control system comprises three sets of coils, each with distinct sizes. Specifically, the coil radii are 20 cm, 14 cm, and 8 cm, respectively. Each pair of coils consists of two identical coils positioned at a separation distance equal to their individual radii. To rectify this, we have included visual representations of the magnetic control system, including images and schematics, in the Supplementary Figure S15. Additionally, we've updated the relevant description in ‘Electromagnetic coils set up’ section (Main text Page 15) to accurately portray the configuration of the coils.

Figure R12. Helmholtz coils are used to generate a uniform magnetic field.

9. There are several mentions of a "beating wave": as they undergo head precession and tail rotation, beating waves are likely not generated at all. Swimming is likely achieved by helical propulsion.

Response: The experimental results suggest that both beating waves and helical propulsion exist in our microswimmers. The behavior is nuanced: the rigid, less deformable helical head predominantly exhibits helical propulsion when actuated by the external magnetic field. However, in the slender tail section, the rotation of the head induces the generation and transmission of beating waves. This coexistence is substantiated by a comparison of the dimensions and shapes of the slender tail under static and dynamic conditions. As shown in the Figure. R13, compared with the static condition, an obvious beating wave shape is observed at the tail of the microrobot and the length is shortened by 19%, which

proves that the tail deforms in the dynamic state rather than simply helical propulsion of head. Please refer to Maintext Page 9 Line 21 and Supplementary Figure 8 in the manuscript.

Figure. R13. Shapes of the slender tail under static and dynamic conditions

10. Page 8: 14.4 mm should read 14.4 microns; by "circle" do you mean "cycle"?

Response: Thank you for your careful review. We have duly amended the measurement from "14.4 mm" to the accurate "14.4 microns" in the manuscript. Regarding the term "circle," if intended as "cycle" in the context you mentioned, we have rectified this as well for precise clarity in the revised version.

11. What do you mean by "average motion straightness"? How is it calculated?

Response: We regret the confusion in our explanation. The term "average motion straightness" refers to the calculation of motion straightness, which is defined as the ratio of VSL (straight-line velocity) to VAP (average path velocity). To derive the average motion straightness, we calculate this ratio for each robot's trajectory and then compute the average of these parameters. Specifically: Motion Straightness = VSL / VAP, where: VSL represents the straight-line velocity, VAP denotes the average path of the robot divided by the total tracking time. This metric of motion straightness helps in characterizing the overall motion patterns of the robots under study.

We have provided related definition in "Average motion straightness calculation" section (Main text, Page 15).

12. Please clarify how the alginate-chitosan bilayer shell forms. Also, why are three layers needed?

Response: Thank you for your inquiry. The formation of the alginate-chitosan bilayer shell arises from the interaction between the numerous primary amino groups (NH^{3+}) on the chitosan molecular chain and the corresponding carboxyl groups (COO^-) on the sodium alginate molecular chain. This interaction leads to the formation of a polyelectrolyte membrane, creating the alginate-chitosan bilayer shell. Further details on this process can be found in the caption of 'Figure 4b' on page 25, line 6 of our manuscript.

Reason for using three layers: The three-layer microcapsule design own several advantages: (1) enhanced stability (2) selectively allow the passage (3) efficient transport of nutrients. This kind of technique has been widely adopted in the biomedical drug/cell

delivery field⁸⁻¹⁰. Therefore, in this manuscript, we have used three-layer membrane structure to increase the structural stability of microswimmers, and to extend the drug release time.

We have discussed this in Maintext. Please find Page 11, line 23 for details.

13. Can you provide any evidence of liquefaction of the alginate core in alkaline environments? And which is the reason for liquefying the core?

Response: In this revised version, we have provided the liquefaction evidence in the Supplementary video 4. It clearly exhibits the fluorescently labeled sol-gel flowing from the microswimmer's core, indicating the transformation of the solid-state gel within the microstructure into a sol-state gel with an aqueous lumen. The reason for liquefying is to demonstrate the process possibility of drug release from the microswimmer's alginate core to the external environment.

We have provided video in this revised version, please refer to Maintext Page 11, line 23.

14. "Sustainable" and "sustain" release are mentioned: did you mean "sustained release"? This is not clearly characterized (the fact the release curve flatten does not imply that all drug molecules will be released at a controlled rate).

Response: Thank you for your attention to this aspect, and we apologize for the lack of clarity in our previous description. To clarify:

1. "Sustained-release" denotes a formulation intended to extend a drug's effectiveness by continuously releasing it within the body over an extended period. This process ensures a slow and non-constant release of the drug, contributing to its prolonged efficacy.

2. "Controlled release" refers to a formulation designed to release the drug at a consistent rate within a specific timeframe. This controlled release mechanism aims to maintain the drug's concentration within the effective range for an extended duration. This can be achieved through a slow and steady release or by closely mimicking a constant release pattern.

We have modified the description of sustained release and revised the description of 'sustained release' and 'release results' in the maintext, please refer to "Gel material enabled pH-sensitive sustained release of microswimmers" section (Page 11 line 12) and (Page 12 line 21).

15. Are magnetic nanoparticles loaded into the realising microswimmers? Please explain this in more detail.

Response: Yes, magnetic nanoparticles are essential components incorporated into the dispersed phase solution (consisting of 1 wt% alginate solution mixed with Fe₃O₄ nanoparticles at 20 wt% with an average size of 230 nm) for fabricating the microswimmers. Please refer to the "Design and fabrication of PSMs" section (Page 4, line 3), Method (Page 14, line 27), and Figure 1a for further details regarding the incorporation of magnetic nanoparticles into the microswimmers.

16. What do the low and high pH represent from a physiological point of view? In which applications can you encounter these extremes? Also, there seems to be very little difference between pH = 7.4 (physiological) and 6.5 (tumour micro-environment, inflamed tissues).

Response: We appreciate your thoughtful inquiry. The choice to assess microswimmer stability under varying pH conditions aimed to simulate environments akin to different regions within the human body. Physiologically, low pH typically represents the acidic environment of the stomach, whereas high pH refers to the more alkaline conditions found in the intestines. The stomach environment is notably acidic (with a low pH), approximately around 1.2, due to the presence of gastric acid. On the other hand, the intestine, particularly the duodenum and small intestine, exhibits different pH levels. Considering this variation, we have selected pH values of 6.5 and 7.1 to represent these environments, respectively. Our study suggests that the microswimmers displayed relatively stable behavior in an acidic environment (pH=6.5) but exhibited reduced stability and limited sustained release capabilities upon exposure to a more neutral to alkaline environment (pH=7.4).

These findings underscore the potential of our microswimmers in biomedical drug delivery. For instance, envisioning potential applications, our microswimmers might be orally administered into the human digestive tract. This delivery approach involves passing through the acidic stomach environment (pH~1.2) and progressing into the small intestine containing alkaline fluids (pH≥6.1) for targeted drug release, illustrating their prospective role in targeted drug delivery within the human body.

We have discussed this in ‘Gel material enabled pH-sensitive sustained release of microswimmers section’ (Page 12 line 14).

17. What is the swelling behaviour reported in Suppl. Fig. 10? Please provide details on these experiments.

Response: The "swelling behavior" mentioned in Suppl. Fig. 10 is an inherent characteristic of hydrogels, whereby the Polymeric Surface Modified Microswimmers exhibit enlargement due to solvent penetration into the void space within the polymeric chain network during the ACA coating process. We notice that as the coating time increased during the encapsulation of microswimmers with ACA layers, the microswimmers expanded in volume, subsequently hindering their swimming capabilities. Suppl. Fig. 10 displays the swelling behavior of ACA-coated PSMs with varying membrane thicknesses (ranging from 2 µm to 10 µm) showcasing a correlation that the microcapsules' swelling degree decreased as the microcapsule membrane's thickness increased.

The method of calculating the swelling degree is previously detailed in our published paper¹¹. In this study, we have followed the same materials and experimental procedures outlined in that publication to encapsulate the PSMs using ACA coating. Please refer to 'Supplementary Section S7 (Page 15, line 6)' and 'Main Text (Page 10, line 25)' for more comprehensive details on this aspect.

18. A proper comparison of the proposed process and micro-swimmers with those used for developing other sperm-like microrobots should be provided.

Response: We appreciate your suggestions. In this revised version, we have introduced a comparative table (Table R1) that delineates the differences between sperm-like microswimmers in terms of fabrication methods, dimensions, and propulsion mechanisms. This table serves to illustrate the distinct features and highlights the unique attributes of our technology. Specifically, our technology excels in processing high-volume, polymorphic sperm robots and implementing efficient actuation strategies. We believe this comparative analysis will offer valuable insights into the distinctiveness and innovative aspects of our proposed microswimmer development. We have added these data in Maintext Table 1.

Table R1. The comparison of sperm-like microswimmers

Author	Fabrication methods	Robot size	Morphology	Propulsion methods	Ref
Ming You, et. al	Magnetic assembly + in-situ polymerization	30 μm	R-head	BW	1
Veronika Magdanz, et. al	Electrostatic self-assembly	34 μm	R-head	HP+BW	2
Friedrich Striggow, et. al	Biological template + Lithography	70 μm	R-I-head	BP	3
Haifeng Xu, et. al	Biological template + Lithography	80 μm	I-head	BW	4
Mariana Medina-Sanchez, et. al	Biological template	50 μm	R-head	HP	5
Islam S. M. Khalil, et. al	Lithography	322 μm	R-head	BW	6
Veronika Magdanz, et. al	Biological template	85 μm	I-head	BP	7
This work	Microfluidic chip	80-120 μm	R-H-I-head	HP+BW	N.A.

*R-head means regular head, I-head means irregular head, H-head means helical head;

**BW means beating waves, BP means biological propulsion, HP means helical propulsion.

Reviewer #3:

The main novelty of this manuscript is that the authors report a method to fabricate alginate structures with elongated "tails" that, when rotated by a magnetic field, can be propelled. The method requires only one-step in a microfluidic device attached to a vortex flow. The fabricated structures can take advantage of the controllable release of substances available to alginate microspheres. In my opinion this manuscript does not rise to the level of significance to be published in Nature Communications. The authors report the new fabrication method, but that is about all. There is no new knowledge gained by using the new devices in an experiment, or new application areas that the devices are used for. This is detailed below.

Response: We extend our gratitude for your thorough assessment of our manuscript.

Fabricating sperm-like microswimmers with effective propelling, desirable loading, and adaptive releasing abilities all in one is of great interest and significance but remains a big challenge. In this manuscript, we present the method of fabricating the nonsymmetric microstructure via vortex turbulence-assisted microfluidics for the first time. Such proposed vortex turbulence-assisted microfluidics approach not only enables the fabrication of polymorphous-tailed microswimmers, but also being a general method for other sophisticated polymorphous structures fabrication that isn't achievable by conventional laminar flow.

As to the lack of depth, in this revised version, we have (1) restructured the qualitative theoretical analysis and simulation section to reveal the mechanism of polymorphous-tailed microswimmer formation; (2) undertaken a comparative analysis between our experimental observations and theoretical predictions; (3) extended the working depth by providing detailed simulation parameters and outlining the magnetic field settings; and (4) conducted more experiments and simulations to address the concerns raised, ensuring a more structured and informative presentation of our contributions.

We hope these revisions can offer a deeper insight into the novelty and significance of our proposed fabrication method. The detailed point-by-point responses to your comments are provided below for your review.

1. The authors do provide some simulations to try to provide "physical principles" of their fabrication method. However, I did not see much explanation or understanding gained from these simulations.

Response: We genuinely appreciate your valuable feedback. The tail formation of the microswimmer can be divided into three processes, i.e., (1) vortex flow generation, (2) droplet deformation, and (3) demulsification. To elucidate the principle and find out the key factors determining these processes, we conduct research through theoretical analysis, simulations, and experiments, where the simulation is crucial. Briefly: The purpose and significance of the simulations in each step are explained below:

(1) Vortex flow generation: We established a mathematic model to investigate this process, which suggests that the shear velocity is the critical parameter (Supplementary Section 2 (Page 5)). Yet, it's difficult to calculate the shear velocity at the demulsification position due to the complex parameters in the model. Therefore, we conduct simulation to

investigate the range of the shear velocity. Our simulation suggests that when the rotation speed of magnetic stirrer increases from 800-1000 rpm, the shear velocity increases from 0.6 m/s to 1 m/s at the demulsification position in the vortex flow. These values are important for the following droplet deformation modeling and simulation analysis.

(2) Droplet deformation: In this process, one key challenge is to figure out the droplet deformation in the vortex flow and the droplet burst state, which is also hard to obtain analytical solutions through modeling calculations. Therefore, we further delved into the mechanics of the droplet during the large deformation and burst process of the alginate-oil capsule through the simulation. Our simulation suggests that when the shear velocity reach the 0.6 m/s (corresponding to magnetic stirrer rotation speeds of 800rpm), the droplet would burst and generate a tiny tail. These values are important for the following demulsification analysis.

(3) Demulsification: After obtaining the trigger value of shear velocity that can generate the microswimmer with tail structure, we need to figure out the polymorphous-tailed microswimmer generation progress, where the transient demulsification process of hydrogel-oil droplets is the key challenge. To address this, we conducted simulation using a 50 μm diameter sphere-shaped alginate droplet with 0.5 μm oil thickness. This approach allowed us to closely examine the evolving demulsification process, observing the behavior of hydrogel droplets in conjunction with the oil flow. Throughout the simulation, we tracked the shaping of hydrogel droplets at the oil-water interface and the subsequent breakage of the oil film triggered by the impact of the vortex flow. Notably, the observed outcomes consistently matched our theoretical analyses and experimental observations, providing valuable insights into the intricacies of the transient demulsification process of hydrogel-oil droplets.

In response to your guidance, we have restructured our article, placing a stronger emphasis on the explanatory aspects of these simulations and the understanding gleaned from our refined theoretical modeling. Please refer to 'Supplementary Section 2 and 3, 5 and 'Main Text (Page 7)' for more comprehensive details.

2. Mostly, the meat of what was actually applied was an analysis of the relevant dimensionless parameters, then a variation of them (by the vortex flow rotation speed and viscosity) to show in what parameter ranges they find different fabricated morphologies. The simulations were not actually necessary for this type of parameter-space exploration. This type of parameter space exploration doesn't really provide any physical insight either.

Response: We truly value your professional insight into our methodology. Different from traditional microfluidic assisted fabrication approaches, there is no referable theory or mechanism for the vortex turbulence-assisted microfluidics fabrication. As explained in Q1, to get a rough image of the overall process, therefore, we employed mathematic model and simulation to assess the parameter-space exploration for discovering the influence of shear velocity on the fabrication process of microswimmers.

As the results shown in Fig. R14, the simulations agree well with the experimental results, affirming the influence of shear velocity on the fabrication process of microswimmers and providing a comprehensive assessment of the parameter-space exploration. This correlation

serves as a valuable predictive tool, enabling us to reduce the workload associated with experiments while fostering a reliable anticipation of processing outcomes. For instance, when aiming to fabricate a new structure, the theoretical and simulation analyses empower us to delineate the relevant parameter ranges. This, in turn, facilitates an efficient fine-tuning of experimental parameters, starting from initial values derived from simulations.

We have discussed this in new section (Maintext ‘Effect of flow speed/ Ca²⁺ concentration on polymorphous tailed-magnetic microswimmers formation’ Page 7).

Figure R14. The comparison of regular microswimmer’s morphology between experimental and simulated conditions at rotational speeds ranging from 800 to 100 rpm.

3. The simulations themselves (described mostly in SM) were not very illuminating. As the authors say in p8 of the SM, their results for the helix head were not observed in the model.

Response: Thank you for your thorough review. As detailed in Q1, the simulation can help us to (1) investigate the effect of the shear velocity on the generated structure of the microswimmers; (2) provide a comprehensive assessment of the parameter-space exploration; and (3) affirm the influence of shear velocity on the fabrication process of microswimmers. Regarding the simulation pertaining to the helix head, our primary objective is to investigate the shear velocity required to initiate the demulsification process and generate polymorphous-tailed structures. The results indicated that at a shear velocity of 0.6 m/s (corresponding to a magnetic stirrer rotation speed of 800 rpm), the droplet would burst, resulting in the formation of a small tail. While the simulation did not replicate the helix head structure due to the inherent complexity of turbulent environments, we emphasize that this limitation does not diminish the overall significance of the simulation in understanding the fabrication process trends.

We acknowledge the importance of a more precise simulation to unravel the details of polymorphous-tailed microswimmer generation. Your feedback underscores the ongoing refinement needed in our simulation methodologies, especially in capturing intricate

structures within turbulent environments. We are committed to addressing these challenges and will incorporate your suggestions into our future work, aiming for a more comprehensive understanding of microswimmer fabrication.

To further enhance our discussion, we have expanded the ‘The formation of polymorphous tailed-magnetic microswimmers’ and ‘Effect of flow speed/ Ca^{2+} concentration on polymorphous tailed-magnetic microswimmers formation’ section in response to your insights. For more detailed information, please refer to maintext Page 7-8 for more details.

4. Also, some of the writing was unclear and not many results were presented of the simulations except for a couple examples. For example, in the sentence after Eq S23, is that a statement about the simulation results, or a statement about the experiments? It would help if maybe the authors were referring to some specific (experimental or simulation) results, but there don't appear to be anything supporting that statement in the simulation results presented.

Response: We deeply appreciate your constructive feedback. Following your kind suggestions, we have seriously revised the content, particularly in the theoretical analysis section, where we referenced specific experimental and simulation results to substantiate our statements more explicitly.

Furthermore, to provide a clearer supporting experimental results of supplementary section S4, we have presented more results in how different shear velocity/ Ca^{2+} influence the generation of the polymorphous tailed microswimmers.

Please check maintext Page 7-8 for more details. In the supplementary section S1-S3, the results refer to the simulations. In the supplementary section S4-S5, the results refer to the experiments.

Figure R15. As the rotational speed increases from 800 to 1000 rpm, the regular microswimmer's body undergoes gradual elongation and exhibits the emergence of a tail-like structure. The morphology of microswimmers generated by the VTAM platform under different calcium ion concentrations.

5. *The method doesn't seem really controllable to make other shapes and applications possible, at least based on what the authors have said. They don't provide an examples of these, for instance. This is made worse since there is not much physical insight into the fabrication method.*

Response: We appreciate your insightful comments. To demonstrate the controllable fabrication of such polymorphous-tailed microswimmers, we have presented the mass production results in this revised version (Figure R16). Briefly, a notably high production yield of helix head microswimmers at an optimal rotation speed of 800 rpm and a 1.5% calcium chloride solution concentration. The regular microswimmers will be massively produced by elevating the rotational speed to 1000 rpm and adjusting the calcium ion concentration to 2%, as evident in Figure R16(iv). We have provided a related description and results in ‘The formation of polymorphous tailed-magnetic microswimmers’ section (Page 6 line 27).

Frankly, we’d like to say the fabrication of the unsymmetrical microswimmers is not as stable as the conventional fabrication of microdroplets at the current stage, caused by the inherent complexity of the turbulent environment. Yet, we believe our primary outcome in this manuscript, that using vortex turbulence assisted microfluidics to fabricate sperm-like microswimmer with effective propelling, desirable loading and adaptive releasing abilities all in one, would inspire the following researchers to develop other sophisticated polymorphous structures fabrication that cannot be achieved by conventional laminar flow. Our ongoing studies will explore and refine processing methods for achieving more precise and controllable fabrication. We have discussed this in Conclusion (Page 14, line 7).

Figure R16. The production of polymorphous sperm-like microswimmers (PSMs).

6. -The numerical calculations of swimming are standard but not really well-described. For example, how is the helical shape of the flexible tail under actuation determined? Is it visualized during rotation? Or do the authors try a large number of possibilities? Since due to flexibility the shape is likely different under rotation than when still, it is not clear how these are accurately estimated.

Response: In the numerical calculation, the parameters, including total length, helix radius, and average pitch angle, are obtained through sample observations by microscope (Dino Lite) in both static and dynamic states. On the basis of a large number of observation, the meanvalues of relevant parameters are obtained for numerical calculations (Supplementary table S1). The parameter that cannot be directly imaged by microscope, like the helical shape of the flexible tail under rotation, are determine by mathematical simulations based on the observable parameters. The external parameters, like the magnetic strength, actuation frequency, liquid viscosity, etc, are set up manually, which are 10 mT, 2 Hz and 8.90×10^{-4} Pa·s respectively.

Despite there might be some deviations from the real value, our calculation shows a high consistency with the experiment in terms of the ranking of the motion performance and the trend of head-tail distribution. In the future, we will try to use high-speed cameras to collect dynamic motion images and obtain more accurate motion parameters for numerical simulation through segmented fitting and finite elements.

In this revised version, we have listed all parameters in a table (Supporting table S1). Related descriptions have been provided in the main text (Page 10) and Supplementary materials (Supplementary Section 7).

7. minor: -In Introduction 2nd paragraph, 1st sentence. The cell membrane is a not a thin rim of cytoplasm.

Response: Thank you for pointing that out. We have revised the sentence in the Introduction's second paragraph as per your suggestion. Instead of "a thin rim of cytoplasm," we have incorporated a more accurate description (Page 15, line 6).

8. -3rd and 4th lines of p3: microspheres are suitable for propulsion near boundaries to fluids, as has been demonstrated in an extensive body of literature. They are also suitable for propulsion in non-Newtonian fluids.

Response: We fully agree with you that microspheres can be driven at fluid boundaries or within non-Newtonian fluids under certain conditions (such as swarm). We regret for our unclear explanation in our manuscript. What we want to express is that "the swimming efficiency and controllability of a completely symmetrical structure similar to a sphere in a liquid environment is much lower than that of a flagellum-like swing drive". In this revised version, we have modified the relevant descriptions to avoid ambiguity among readers. Please check in maintext Page3, line 3.

References:

1. G. Kaklamani, D. Cheneler, L. M. Grover, M. J. Adams, J. Bowen, Mechanical properties of alginate hydrogels manufactured using external gelation. *J Mech Behav Biomed Mater* **36**, 135-142 (2014).
2. T. Ramdhan, S. H. Ching, S. Prakash, B. Bhandari, Time dependent gelling properties of cuboid alginate gels made by external gelation method: Effects of alginate-CaCl₂ solution ratios and pH. *Food Hydrocoll.* **90**, 232-240 (2019).
3. E. Mammarella, D. De Piante Vicin, A. Rubiolo, Evaluation of stress-strain for characterization of the rheological behavior of alginate and carrageenan gels. *Brazilian Chem. Eng. J.* **19**, 403-409 (2002).
4. A. Saarai, V. Kasparikova, T. Sedlacek, P. Sáha, On the development and characterisation of crosslinked sodium alginate/gelatine hydrogels. *J Mech Behav Biomed Mater* **18**, 152-166 (2013).
5. A. Tiribocchi *et al.*, The crucial role of adhesion in the transmigration of active droplets through interstitial orifices. *Nat. Commun.* **14**, 1096 (2023).
6. Y. Zhou, C. Zhang, W. Zhao, S. Wang, P. Zhu, Suppression of hollow droplet rebound on super-repellent surfaces. *Nat. Commun.* **14**, 5386 (2023).
7. Z. Yuan *et al.*, Ultrasonic tweezer for multifunctional droplet manipulation. *Sci. Adv.* **9**, eadg2352 (2023).
8. J.-F. Zhang, Z.-Q. Liu, X.-H. Zhang, Y.-G. Zheng, Biotransformation of iminodiacetonitrile to iminodiacetic acid by *Alcaligenes faecalis* cells immobilized in ACA-membrane liquid-core capsules. *CHEM PAP* **68**, 53-64 (2014).
9. P. Ylivero, C. J. Franzén, M. J. Taherzadeh, Mechanically robust polysiloxane-ACA capsules for prolonged ethanol production. *J. Biotechnol.* **88**, 1080-1088 (2013).
10. Y. Liu *et al.*, Programmable higher-order biofabrication of self-locking microencapsulation. *Biofabrication* **11**, 035019 (2019).
11. Y. Liu *et al.*, Multi-functionalized micro-helical capsule robots with superior loading and releasing capabilities. *J Mater Chem B* **9**, 1441-1451 (2021).

REVIEWER COMMENTS

Reviewer #1 (Remarks to the Author):

In this revised version, the author made a significant improvement in the revised manuscript by clarifying their contributions in detail and fairly addressed all previous questions with additional new experiments. Compared with the previous version, the comparison of theoretical and experimental results is much clearer. The newly added experiments fully prove and reveal the effects of rotation speed and calcium ion concentration on the microswimmers's formation. Particularly, they provided conclusive evidence supporting the production of polymorphous microswimmers. I suggest it for publication in NC and wish it can bring new inspirations to the readers.

Some minor comments related to the manuscript:

- 1) The legend b in Figure 3 describes Figure 3c and needs to be corrected.
- 2) Please double check the references and make sure all related references have been included.
- 3) The scale bar in irregular head microswimmes (Figr 2d) is missing.
- 4) On Page 5 Para 2: "extracted by the votrex flow" should be "extracted by the vortex flow"

Reviewer #2 (Remarks to the Author):

Thanks for the in depth amendment of the manuscript. Below are my comments to the individual responses.

1. OK

2. OK

3. OK

4. My comment was not about 2D vs. 3D geometry. I am fully aware that reducing the numerical complexity with 2D simulations can lead to much faster and even more accurate solutions. My comment was about the shape of the simulated domain (wide and shallow) vs. the shape of the vial. Also, were these 2D planar or 2D axisymmetric simulations? Please provide details on the physics, models, and boundary conditions adopted.

5. Please provide details of how density and viscosity were measured. Also, the viscosity of the CaCl₂ solution seem unrealistically high (more than 3x that of water).

6. What does "surface concentration" mean?

7. The reported pH of DI water is slightly lower than the usual range of DI water.

8. OK

9. What I meant by "helical propulsion" is that the tail likely wind into an helical shape rather than beating by undulatory waves (which is commonly achieved by oscillating rather than precessing magnetic fields). I did no refer to the propulsion by he head. The observed shortening of the tail could be explained by either beating or winding deformation and it does not prove that undulatory beating occurs.

10. OK

11. OK, it is clear now.

12. OK

13. Alginate liquefies in presence of phosphate and citrate as they chelate Ca²⁺ ions. Both substances were used in the reported pH 10-12 dissolution tests. The observed results might be due to the presence of phosphate and citrate rather than by the pH. The results should be confirmed by additional tests in a strong base solution (e.g. NaOH), rather than in a phosphate/citrate buffer. Moreover, information on which buffers were used for the pH 1.2, 6.8 and 7.4 dissolution tests is missing.

14. OK

15. OK

16. OK

17. OK

18. Thank you for this detailed comparison. However I feel that such a table is not needed. Rather, what is needed is a critical discussion on advantages and drawbacks of the proposed fabrication method compared to those in the literature.

Response to reviews for the manuscript

We would like to thank all reviewers for thoroughly reading our manuscript entitled **“One-step formation of polymorphous sperm-like microswimmers by vortex turbulence assisted microfluidics”**. Those comments are helpful for raising the quality of the paper, as well as with the important guiding significance to our research.

In this response letter, we answered the comments in a point-by-point manner. For the convenience of the reviewers, the comments and suggestions are listed below in the *blue font*, followed by our responses in the normal black font. We also highlighted the corresponding modification in our revised manuscript in the *red font*.

Reviewer #1:

In this revised version, the author made a significant improvement in the revised manuscript by clarifying their contributions in detail and fairly addressed all previous questions with additional new experiments. Compared with the previous version, the comparison of theoretical and experimental results is much clearer. The newly added experiments fully prove and reveal the effects of rotation speed and calcium ion concentration on the microswimmers's formation. Particularly, they provided conclusive evidence supporting the production of polymorphous microswimmers. I suggest it for publication in NC and wish it can bring new inspirations to the readers.

Response: Thanks for the reviewer's kind recommendation.

1. The legend b in Figure 3 describes Figure 3c and needs to be corrected.

Response: Thanks for the reviewer's kind reminder. We have revised this legend of Figure 3. Please check in Maintext Page 27 the legend of Figure 3.

2. Please double check the references and make sure all related references have been included.

Response: Thanks for the reviewer's kind and careful reminder. We have edited the references cited in supplementary form and moved them to the end of the reference list to make it easier for reading. Please check in the Maintext Page 19.

3. The scale bar in irregular head microswimmers (Firm 2d) is missing.

Response: Thanks for the reviewer's kind reminder. We have added the scale bar and verified the Figure 2. Please check in the Maintext Page 22.

4. On Page 5 Para 2: "extracted by the vortex flow" should be "extracted by the vortex flow"

Response: Thanks for the reviewer's kind reminder. We have revised this typo and checked the whole of the manuscript again. Please check in the the Maintext Page 4 line 27.

Reviewer #2:

Thanks for the in depth amendment of the manuscript. Below are my comments to the individual responses.

Response: We sincerely appreciate the reviewer's comment, which helps us improve our manuscript a lot. We have also carefully considered the remaining comments and suggestions and have made corresponding improvements as presented in this response letter.

1. OK

Response: Thanks for the reviewer's kind approval.

2. OK

Response: Thanks for the reviewer's kind approval.

3. OK

Response: Thanks for the reviewer's kind approval.

4. My comment was not about 2D vs. 3D geometry. I am fully aware that reducing the numerical complexity with 2D simulations can lead to much faster and even more accurate solutions. My comment was about the shape of the simulated domain (wide and shallow) vs. the shape of the vial. Also, were these 2D planar or 2D axisymmetric simulations? Please provide details on the physics, models, and boundary conditions adopted.

Response: Thanks for your comments. We have conducted the simulations in two different setups to estimate the effect of shape and dimension. One setup is 10 mm wide and 25 mm shallow same as the vial, while the other one is 3 mm wide and 0.3 mm shallow. As shown in Fig. R1 and Fig. R2, the simulation results in these two setups are similar. Considering the latter one can greatly reduce the simulation time (from 4 hours to 10 mins), we choose the latter setup for simulation in this manuscript. The experimental observation showcases that the demulsification and deformation process is confined to micrometer size and milli-seconds, which is orders of magnitude smaller than the dimension of the simulation environment. Therefore, the shape of the simulated domain and the shape of the vial would not influence the results significantly.

Figure R1. The simulation of experimental vial shape.

Figure R2. The oversimplified simulation of this paper

The simulation environment we employ in this manuscript is 2D planar. The reason is that our goal is to investigate the influence of shear velocity on the fabrication process of microswimmers, where 2D planar simulation can take all key factors into consideration while maintaining a low time and computing resource consumption. Therefore, 2D planar simulation was adopted in our work. In future work, we will consider implementing more complex simulations.

The detailed parameters and boundary conditions are given in Table R1. Briefly, we use “Turbulent Flow” to simulate the flow, and set up two inlet conditions to represent the flow in the bottle (Fig. R3, inlet1) and the flow induced by alginate dropping into the calcium mixture solution (Fig. R3, inlet2). To simulate the demulsification process of the hydrogel-oil droplets, we used “Ternary Phase Field” to couple with the “Turbulent Flow”, while setting up “Three Phase Flow, Phase Field” in multiphysics. In this three-phase flow field, we can simulate the flow of three immiscible fluids separated by a moving interface. To simulate the deformation process of the microswimmer (alginate droplet concentration diffusion process), we use “Transport of Diluted Species” to couple with the “Turbulent Flow”, while setting up “Reacting Flow, Diluted Species” in Multiphysics^{1,2}. After setting the boundaries depicted in Table R1, we finally get the demulsification process of the hydrogel-oil droplets (Maintext Fig. 2b and Supplementary Video 1), and the deformation process of the microswimmer (Maintext Fig. 3a).

The detailed description as supplied in Supplementary Page 7, Section 3 and Page12, Section 5. Please check it.

Table R1

density of alginate, oil, CaCl ₂ mixture	$1.6 \times 10^3 \text{ kg/m}^3$, 800 kg/m^3 and $1000 \sim 1025 \text{ kg/m}^3$, respectively
viscosity of alginate, oil, CaCl ₂ mixture	0.0137 Pa.s, 0.032 Pa.s, and 0.00325 Pa.s, respectively
inlet1 velocity	0.6-1 m/s, measured by experimental value ^[3]
inlet2 velocity	0.0212-0.0255m/s, measured by experimental value
outlet pressure	0 Pa

alginate phase, defined by ($\phi_{alginate} + \phi_{oil} + \phi_{calcium} = 1$)	$\phi_{alginate} = 1$, $\phi_{oil} = 0$, $\phi_{calcium} = 0$
oil phase, defined by ($\phi_{alginate} + \phi_{oil} + \phi_{calcium} = 1$)	$\phi_{alginate} = 0$, $\phi_{oil} = 1$, $\phi_{calcium} = 0$
calcium solution phase, defined by ($\phi_{alginate} + \phi_{oil} + \phi_{calcium} = 1$)	$\phi_{alginate} = 0$, $\phi_{oil} = 0$, $\phi_{calcium} = 1$, as the boundaries condition of inlet
alginate droplet concentration	$c = 7.4 \text{ mol/m}^3$ ($M_{alg} = 216.121 \text{ g/mol}$), calculated by experimental value
the inlet in "Turbulent Flow"	$c = 0$

Figure R3. The description of boundaries condition.

5. Please provide details of how density and viscosity were measured. Also, the viscosity of the CaCl₂ solution seem unrealistically high (more than 3x that of water).

Response: Thanks for the reviewer's valuable suggestions.

- About density** - To measure the density, we filled the solution in a 1ml PE tube, and then measure the solution mass (m) on a scale (RADWAG AS 220.R2 PLUS, Poland). The density of solutions was calculated by dividing the mass by volume (m/v).
- About viscosity** - The viscosity was measured by viscometers (Techcom SNB-2E) at 25 °C in a thermostatic circulating water bath (DC-0506).
- About the viscosity of the CaCl₂** – We agree with you that the viscosity of pure CaCl₂ solution is similar to that of water. However, in our experiment, the residual portion of the paraffin oil and surfactant solution is also mixed into CaCl₂ solution during the testing process, resulting in the viscosity of the mixed CaCl₂ solution being much higher than the pure CaCl₂ solution. We regret we didn't present this clearly in our previous version. In this revised version, we have corrected it to "CaCl₂ mixture with paraffin oil and surfactant solution" called "CaCl₂ mixture" in short.

We revised the description of calcium solution, and provide detailed characterization process of density and viscosity. Please check Methods in Maintext Page15.

6. What does "surface concentration" mean?

Response: Thanks for the reviewer's comments. The surface concentration means alginate phase concentration in deformation process.

7. The reported pH of DI water is slightly lower than the usual range of DI water.

Response: Thanks for pointing out this issue. The DI water we used is obtained from Milli-Q Advantage A10, whose pH is 7 initially. However, after the DI water is exposed to air, it absorbs CO₂ and forms carbonic acid (H₂CO₃), resulting in a lower pH value altering from 7.0 to 5.5⁴. In this experiment, we measured the pH value of the DI water at the time of conducting the swimming experiment, thereby its value is lower than 7.0. In this manuscript, the pH of DI water is not a parameter affecting the stability and locomotion of the microswimmer, as shown in question 13.

We have revised our description of DI water pH (pH~5.5-7). Please see the Maintext Page 9 line 24.

8. OK

Response: Thanks for reviewer's satisfaction with our response.

9. What I meant by "helical propulsion" is that the tail likely wind into an helical shape rather than beating by undulatory waves (which is commonly achieved by oscillating rather than precessing magnetic fields). I did no refer to the propulsion by he head. The observed shortening of the tail could be explained by either beating or winding deformation and it does not prove that undulatory beating occurs.

Response: We're sorry that our misnomer "beating wave". What we want to express is that the soft tail can form a helical-like shape under the driven of head and provides propulsion force, which is consistent with your point of view.

To eliminate ambiguity, we have deleted the description of "beating wave" in the revised version. Meanwhile, we have also explained it in detail in the dynamic model in the supplementary material (Section S6, page 12), which has also been widely demonstrated in other related literature^{5,6}.

10. OK

Response: Thanks for reviewer's satisfaction with our response.

11. OK, it is clear now.

Response: Thanks for reviewer's affirmative approval.

12. OK

Response: Thanks for reviewer's satisfaction with our response.

13. Alginate liquefies in presence of phosphate and citrate as they chelate Ca^{2+} ions. Both substances were used in the reported pH 10-12 dissolution tests. The observed results might be due to the presence of phosphate and citrate rather than by the pH. The results should be confirmed by additional tests in a strong base solution (e.g. NaOH), rather than in a phosphate/citrate buffer. Moreover, information on which buffers were used for the pH 1.2, 6.8 and 7.4 dissolution tests is missing.

Response: Thank you for your attention to this aspect. Both the chelation (phosphate, citrate, etc.) and pH are key factors for the alginate liquefies. The chelation is necessary for the generation of liquefaction, i.e. the Ca-alginate substantial egg box structure is destroyed by chelate ions and removing Ca^{2+} from alginate inlayer under the ion exchange chelation sequentially. It's worth noting that such a reaction only occurs efficiently in weakly acidic to alkaline environment due to deprotonation^{7,8}. Therefore, both the existence of chelation and a weakly acidic to alkaline pH environment is needed for the liquefaction process.

To verify it, we conducted additional experiments based on your comment: (1) high pH (0.6M sodium hydroxide solution (pH=12-13)) without chelation. (2) chelation with different pH (100 μ l of the same concentration PBS with different pH value (0.1mol/l, pH=3, 5,7, 9, Yuanye company, Shanghai). The results indicate that liquefaction cannot proceed without either ion chelation or weakly acidic to alkaline pH, as shown in Fig. R4.

The buffer we used is 0.1N HCl (pH1.2) and PBS (phosphate buffer solution, 0.1mol/l, pH=6.8 and pH=7.4) in the test, considering the following two reasons: (1) the main component of gastric acid in the human digestive juice environment was hydrochloric acid⁹; and (2) the intestinal juice contains various phosphate ions, bicarbonate ions, sodium ions, potassium ions, etc¹⁰.

We have provided related descriptions and results in the “Gel material enabled pH-sensitive sustained release of microswimmers” section (Page 11, line 20) and Supplementary Figure S11.

Figure R4. The liquefaction process both affected by chelating ions and pH.

14. OK

Response: Thanks for the reviewer's kind approval.

15. OK

Response: Thank the reviewer for being satisfied with the response.

16. OK

Response: Thanks for the reviewer's kind approval.

17. OK

Response: Thank the reviewer for being satisfied with the response.

18. *Thank you for this detailed comparison. However I feel that such a table is not needed. Rather, what is needed is a critical discussion on advantages and drawbacks of the proposed fabrication method compared to those in the literature.*

Response: Thank you for your suggestion. In the revised version, we have removed the table and discussed the advantages and drawbacks of our fabrication methods in the discussion section, briefly:

“Different to the manufacture of traditional microswimmers¹¹⁻¹⁷, the proposed vortex turbulence assisted microfluidics (VTAM) approach enables the fabrication of biocompatible asymmetrical magnetic microswimmers with tail in one step. It not only provides high moveable ability but also achieves bionic semi-permeable membrane encapsulation, offering sustain release capability for targeted drug delivery.” “It's worth mentioning that this manuscript mainly focuses on the proof of concept of the fabrication process. Compared to the conventional microcapsule fabrication approaches, there is a lot of room for improvement, such as fabrication controllability, stability, unity, and productivity. Moreover, further in-vivo tests are also required for the structure optimization of the microswimmer to make it clinically applicable.”

Please check detailed in Maintext Page 13-14.

Ref:

1. N. Oikawa, R. Kurita, One-Way Diffusion and Active Motion of Ionic Liquids in a Dissolution Process in Water. *Complexity and Synergetics*, 167-175 (2018).
2. Y. Wang, Y. Xu, Y. Wang, B. Li, C. Wang, Z. Han, L. Weng, Reaction–Diffusion Process for Hydrogels with a Tailored Layer Structure. *PROCESSES* **11**, 1975 (2023).
3. G. Halász, B. Gyüre, I. M. Jánosi, K. G. Szabó, T. Tél, Vortex flow generated by a magnetic stirrer. *Am. J. Phys.* **75**, 1092-1098 (2007).
4. E. Riché, A. Carrié, N. Andin, S. Mabic, High-purity water and pH. *American laboratory* **38**, 22 (2006).
5. I. S. Khalil, A. F. Tabak, M. Abou Seif, A. Klingner, B. Adel, M. Sitti, Swimming in low Reynolds numbers using planar and helical flagellar waves. *IEEE/RSJ (IROS)*, pp. 1907-1912

(2017).

6. I. S. Khalil, A. F. Tabak, M. Abou Seif, A. Klingner, M. Sitti, Controllable switching between planar and helical flagellar swimming of a soft robotic sperm. *PLoS one* **13**, e0206456 (2018).
7. L. Lacerda, A. L. Parize, V. Favere, M. C. M. Laranjeira, H. K. Stulzer, Development and evaluation of pH-sensitive sodium alginate/chitosan microparticles containing the antituberculosis drug rifampicin. *Mat Sci Eng C-mater* **39**, 161-167 (2014).
8. Q. Zhao, B. Li, pH-controlled drug loading and release from biodegradable microcapsules. *Nanomedicine: Nanotechnology, Biology and Medicine* **4**, 302-310 (2008).
9. T. C. Martinsen, K. Bergh, H. L. Waldum, Gastric juice: a barrier against infectious diseases. *BASIC CLIN PHARMACOL* **96**, 94-102 (2005).
10. J. S. Fordtran, T. W. Locklear, Ionic constituents and osmolality of gastric and small-intestinal fluids after eating. *AM J GASTROENTEROL* **11**, 503-521 (1966).
11. M. You, F. Mou, K. Wang, J. Guan, Tadpole-Like Flexible Microswimmers with the Head and Tail Both Magnetic. *Appl. Mater. Interfaces* **15**, 40855-40863 (2023).
12. V. Magdanz, I. S. Khalil, J. Simmchen, G. P. Furtado, S. Mohanty, J. Gebauer, H. Xu, A. Klingner, A. Aziz, M. Medina-Sánchez, IRONSperm: Sperm-templated soft magnetic microrobots. *Sci. Adv.* **6**, eaba5855 (2020).
13. F. Striggow, M. Medina-Sánchez, G. K. Auernhammer, V. Magdanz, B. M. Friedrich, O. G. Schmidt, Sperm-driven micromotors moving in oviduct fluid and viscoelastic media. *Small* **16**, 2000213 (2020).
14. H. Xu, M. Medina-Sánchez, V. Magdanz, L. Schwarz, F. Hebenstreit, O. G. Schmidt, Sperm-hybrid micromotor for targeted drug delivery. *ACS nano* **12**, 327-337 (2018).
15. M. Medina-Sánchez, L. Schwarz, A. K. Meyer, F. Hebenstreit, O. G. Schmidt, Cellular cargo delivery: Toward assisted fertilization by sperm-carrying micromotors. *Nano Lett.* **16**, 555-561 (2016).
16. I. S. Khalil, H. C. Dijkslag, L. Abelmann, S. Misra, MagnetoSperm: A microrobot that navigates using weak magnetic fields. *Appl. Phys. Lett.* **104**, (2014).
17. V. Magdanz, S. Sanchez, O. G. Schmidt, Development of a sperm-flagella driven micro-bio-robot. *Adv. Mater.* **25**, 6581-6588 (2013).

REVIEWERS' COMMENTS

Reviewer #1 (Remarks to the Author):

The authors have addressed my concern, so this reviewer suggest the publication without modifications.

Reviewer #2 (Remarks to the Author):

The authors addressed all previous comments by the reviewer.

Response to reviews for the manuscript

We would like to thank all reviewers for thoroughly reading our manuscript entitled **“One-step formation of polymorphous sperm-like microswimmers by vortex turbulence assisted microfluidics”**.

Reviewer’s comment (*Italic Blue*) and our answer (Normal black):

Reviewer #1:

The authors have addressed my concern, so this reviewer suggest the publication without modifications.

Response: Thanks for the reviewer’s kind recommendation.

Reviewer #2:

The authors addressed all previous comments by the reviewer.

Response: Thanks for the reviewer's kind approval.